# Parallel developmental changes in children's production and recognition of line drawings of visual concepts

Bria Long [1] ✉, Judith E. Fan [1,2], Holly Huey [2], Zixian Chai[1] & Michael C. Frank [2]

Childhood is marked by the rapid accumulation of knowledge and the prolific production of drawings. We conducted a systematic study of how children create and recognize line drawings of visual concepts. We recruited 2-10-year-olds to draw 48 categories via a kiosk at a children's museum, resulting in >37K drawings. We analyze changes in the category-diagnostic information in these drawings using vision algorithms and annotations of object parts. We find developmental gains in children's inclusion of category-diagnostic information that are not reducible to variation in visuomotor control or effort. Moreover, even unrecognizable drawings contain information about the animacy and size of the category children tried to draw. Using guessing games at the same kiosk, we find that children improve across childhood at recognizing each other's line drawings. This work leverages vision algorithms to characterize developmental changes in children's drawings and suggests that these changes reflect refinements in children's internal representations.

What makes a drawing of a rabbit look like a rabbit and not a dog? As adults, our visual concepts—our sense of what particular objects look like—are seamlessly integrated into our visual experience. With a single glance, incoming patterns of light make contact with our visual concepts, supporting the rapid categorization of a wide variety of inputs, from real-life exemplars to sparse line drawings[1–4]. We can also access our visual concepts in the absence of perceptual input—going beyond what we have experienced to imagine and create new visual entities[5,6].

While these feats of perceiving and creating can feel effortless, the representations that support them are acquired gradually as children learn about the visual world[7]. Infants' everyday home experiences scaffold their earliest representations[8], but children begin building visual concepts in earnest as they learn which labels refer to both depictions and real-life exemplars of categories[9]. And by their second birthday, children can learn category labels for novel objects after exposure to just a few exemplars[10,11] and succeed for sparse 3D representations devoid of color and texture-based cues[11].

But children take many years to learn how to appropriately generalize and discriminate between visual concepts. For example, children gradually improve in their ability to accurately group together categories based on taxonomy versus salient perceptual features (e.g., grouping a snake with a lizard vs. a hose)[12,13]. Further, children's visual recognition abilities have a protracted developmental trajectory throughout middle childhood[14,15] as children become steadily better at discriminating between similar exemplars of scenes, objects, bodies, and faces[16], and increasingly skilled at recognizing objects across unusual poses or 3D rotations[14,17,18]. In turn, changes in children's recognition abilities are related to changes in how the visual cortex encodes objects and scenes[17,19–21]; for example, children's ability to discriminate similar faces is correlated with the sensitivity of face-selective regions to these particular faces[22]. These changes in children's ability to discriminate exemplars may be driven by children's increasing attention to the relationships between object parts and their overall configuration[15,23,24]. Together, these findings suggest that visual concepts are refined throughout early and middle childhood as children learn how to discriminate between similar categories.

Psychologists have typically probed children's visual concepts by asking children to make discrete choices between small samples of

[1]Department of Psychology, Stanford University, Building 420, 450 Jane Stanford Way, Stanford 94305 CA, USA. [2]Department of Psychology, University of California, San Diego, 9500 Gilman Dr., La Jolla 92093 CA, USA. ✉e-mail: bria@stanford.edu

stimuli that vary along dimensions chosen by an experimenter. While valuable for testing specific hypotheses, this strategy is also characterized by limits on the amount of information that can be acquired on any given experimental trial. By contrast, generative tasks such as drawing production can overcome these limits by enabling the collection of more information about the contents of children's visual concepts on every trial. Further, almost all children prolifically produce drawings of visual concepts from an early age[25], and there is substantial precedent for the examination of children's drawings to probe their knowledge about the visual world[25–27]. Freehand drawing tasks thus provide a valuable tool for characterizing developmental changes in visual concepts. Here, we create a large digital dataset of children's drawings and leverage innovations in machine learning to characterize how changes in children's drawings are related to their growing understanding of various visual concepts.

Our work builds on a long literature that has argued that children's drawings of objects reflect not only what they can directly observe, but what they know about these objects [see intellectual realism in ref. 28, 29]. For example, even when drawing from observation, children tend to include features that are not visible from their vantage point but are nevertheless diagnostic of category membership (e.g., an occluded handle on a mug)[30,31]. Further, direct visual or haptic experience with novel objects tends to change what information children draw[30]. These initial studies have focused on a small number of visual concepts—especially the human figure[32]—finding that younger children (4–5 years) tend to include fewer category-diagnostic cues, such as cues distinguishing an adult from a child, than somewhat older children (6 years), who tend to enrich their drawings with more diagnostic part information[33,34]. However, the generality of the conclusions based on this work has been unclear given the narrow range of concepts tested and the lack of generic methods for measuring diagnostic information in drawings. Further, little work has systematically related children's ability to include diagnostic visual information in drawings to their emerging abilities to control and plan their motor movements[35,36].

Yet research in adults does suggest that what we draw is tightly linked to what we know about objects and how we perceive them. For example, patients with semantic dementia tend to produce drawings without distinctive visual features[37] or include erroneous features (e.g., a duck with four legs). One recent study found that adults can produce detailed drawings of scenes after only viewing them for a few seconds[38]. Another study found that recognizing an object and producing a drawing of an object recruit a shared neural representation in early visual cortex[39]. Further, practice producing drawings of objects can impact perceptual judgments about them. In one study, adults who repeatedly drew similar objects (i.e., beds vs chairs) were better able to distinguish them in a categorization task[40]. Drawing expertise is also associated with enhanced visual encoding of object parts and their relationships[41–44], but not differences in low-level visual processing[43–46] or shape tracing skills[47].

Building on these traditions, in the current paper we characterize developmental changes in how children produce and recognize line drawings as an additional lens into children's growing understanding of these visual concepts. We anticipated that children's ability to produce and recognize line drawings would continue to develop beyond the preschool and early elementary school years[16,17,20] and that some—but not all—age-related variation in drawing ability would be due to improvements in planning and motor control[35,36]. In particular, as children learn the visual information most diagnostic of a visual concept[7], this visual knowledge may manifest in both: (1) an enhanced ability to produce line drawings that contain category-diagnostic information and (2) a greater sensitivity to this same visual information when recognizing line drawings made by other children.

We thus first collected digital drawings of 48 different visual concepts from a large sample of children spanning a wide age range (2–10 years), resulting in a corpus containing >37K drawings. To quantify developmental changes in these drawings, we leveraged techniques from modern machine learning and computer vision, in particular the latent feature representations learned by large neural networks trained on visual discrimination tasks[48,49], which have been shown in prior work to capture meaningful variation in human perceptual judgments about both natural images and drawings[40,50]. We use these latent feature representations both to quantify the category-diagnostic information in each drawing and to analyze the similarity structure in children's unrecognizable drawings. We then crowd-sourced part labels for each stroke in a subset of these drawings to quantify how the parts children included in their drawings changed across development. Finally, we administered drawing recognition tasks to measure how well children of different ages could identify which visual concept a given drawing was intended to convey.

This study makes a number of contributions relative to the prior literature. First, we collect, annotate, and share a large sample of children's drawings from scribbles to sophisticated sketches, creating valuable resources for future research. Second, we develop an analytic approach suitable for exploring these drawings, which yields a number of intriguing findings around drawing development—including the presence of semantic information even in children's unrecognizable drawings. Finally, we find evidence for the relation between developmental changes in children's drawing abilities and their growing understanding of the visual concepts they are drawing. Older children include more diagnostic visual information and relevant object parts when producing line drawings, and these gains are not easily explainable by category exposure frequency or visuomotor development. Further, children's developing ability to recognize drawings is related to the presence of category-diagnostic information in these drawings. Together, we provide a set of tools and insights into the development of drawings and visual representations in childhood, which we hope will spur future research on this topic.

## Results
### Examining drawing production
We installed a free-standing kiosk at a children's science museum (see Fig. 1a), where children used a touchscreen tablet to produce their drawings. We included a set of shape-tracing trials in the drawing production task to measure children's tracing skills (see Fig. 1b). After completing these tracing trials, children were verbally prompted to draw different visual concepts, including both animals and inanimate objects that are both commonly drawn (e.g., face, cat) and less commonly drawn by children (e.g., octopus, piano) (see *Methods*, Supplementary Fig. 1, Supplementary Fig. 2). After filtering, the dataset contained 37,770 drawings of 48 categories from $N = 8084$ children (average age: 5.33 years old; range: 2–10 years old; see Supplementary Table 1 for age demographics, Supplementary Table 2 for reported interference rates).

Measuring category-diagnostic information in such a large dataset of children's drawings poses a major analytical challenge. Until recently, researchers analyzing even small drawing datasets had to develop ad hoc criteria for scoring drawings based on their intuitions about what the distinctive visual features could be (e.g., handles for mugs)[31,32]. Fortunately, recent advances in computer vision have made it possible to measure category-diagnostic information at scale by leveraging latent feature representations learned by large neural networks[48,49], although at some cost to interpretability, as these learned features are not guaranteed to map onto nameable object parts. We thus use two approaches with complementary strengths: first, we use model classifications to estimate the amount of category-diagnostic information in each drawing; second, we use crowd-sourcing to identify which parts children included in their drawings.

Our first approach leverages the latent feature representations learned by neural network models to derive measures of a drawing's

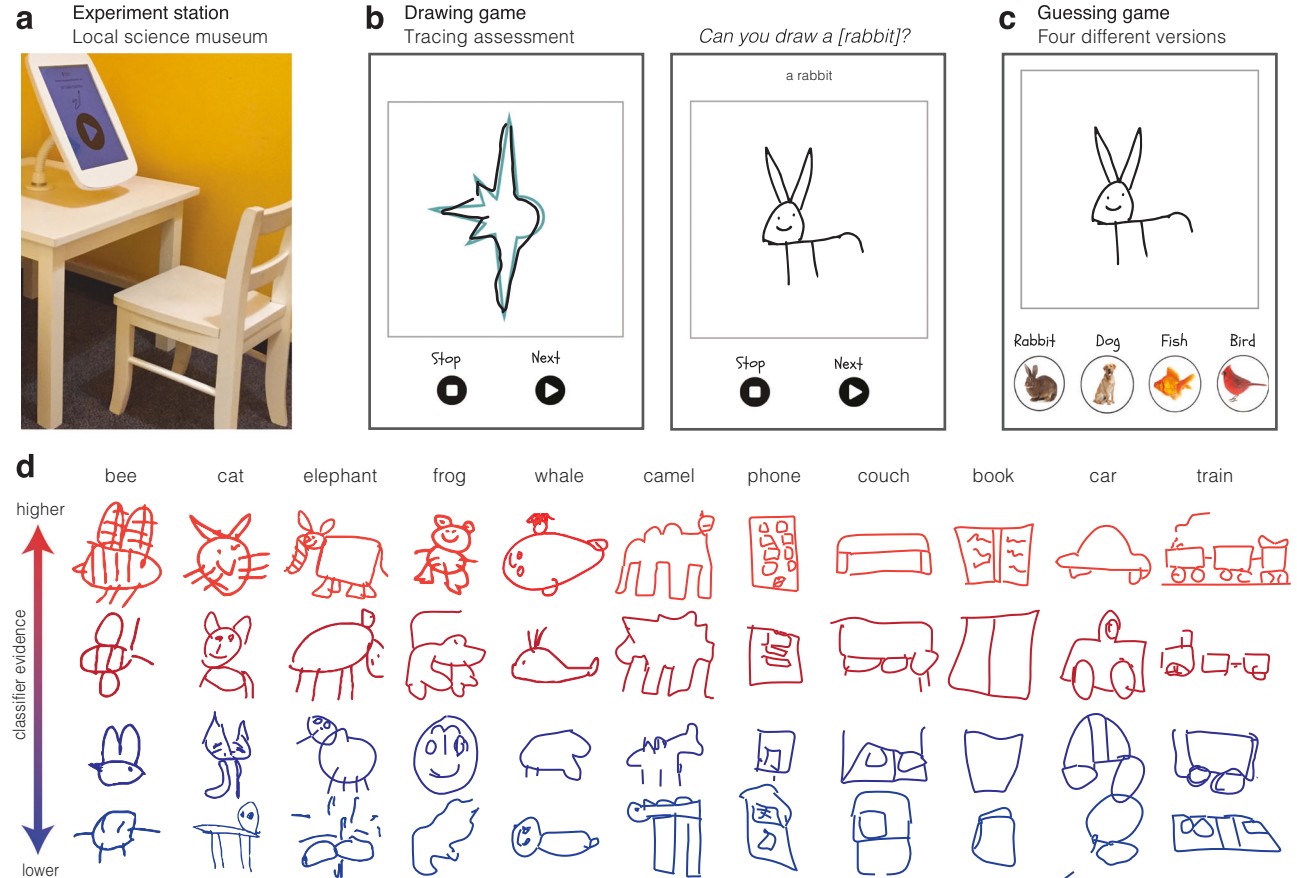

**Fig. 1 | Overview of tasks and example drawings. a** Museum kiosk where children participated, and **b** examples of the tracing, drawing, and **c** guessing trials. **d** Example drawings from several categories; redder drawings contain more diagnostic visual information (as assessed by classifier evidence using VGG-19 FC6 features, see *Methods*).

recognizability—how much category-diagnostic information it contains. Specifically, we analyzed the degree to which the visual features of each drawing could be used to decode the category that children intended to draw. Activations for each sketch were taken from the second-to-last layer of VGG-19, a deep convolutional neural network pre-trained on Imagenet classification[48], as prior work has shown that activations from deeper layers tend to correspond to the visual features that enable basic-level recognition (e.g., cat vs. dog) in both sketches and photographs[40]. These features were then used to train logistic-regression classifiers to predict which of the 48 categories children were asked to draw for sets of held-out drawings (see *Methods*), balanced across categories. For every drawing, this procedure thus yielded: (1) a binary classification score, indicating whether a given drawing contained the visual features that enabled basic-level recognition, and (2) a probability score for each of the 48 categories, capturing the degree to which a given drawing contained the visual features relevant to that category (Fig. 2a, Fig. 1d). We then validated these VGG-19 model classifications by using embeddings from a contrastive language-image pre-training model (CLIP[49]), which jointly trains an image and a text encoder to predict image and text pairings. While relatively less work has related the embeddings of this model class to either human behavioral or neural representations[51], CLIP outperforms many models at recognizing visual concepts across different visual formats[49].

For our second approach, we crowd-sourced human annotators to tag every stroke in a subsampled set of *N* = 2160 drawings with a part label (see *Methods*). Using these annotations, we analyzed changes in which parts children drew and how much they emphasized those parts in their drawings. The goal of these analyses was to provide insight into

which specific elements within children's drawings change across development and give rise to the changes in category-diagnostic information measured using model classifications.

## Drawings of visual concepts become more recognizable across childhood

Children's drawings steadily increased in recognizability with age, as measured using model classification performance (Fig. 2b, fixed effect of age in Table 1, see *Methods*; validation using CLIP in Supplementary Table 3, Supplementary Fig. 3, Supplementary Fig. 4). Features from deeper layers of VGG-19 were critical to recovering these age-related changes, suggesting that drawings produced by older vs. younger children primarily differed in mid- and high-level visual features (Supplementary Fig. 5). We replicated this finding in a separate controlled experiment in which a researcher was present[52], suggesting that these effects were not an artifact of data collection at the kiosk (fixed effect of age in Supplementary Table 4, subset of data in ref. 52, Supplementary Fig. 6).

What explains these gradual increases in recognizability? Younger children may simply have had less practice drawing and are thus less well-equipped to express what they know, despite a mature understanding of these categories. This account predicts that changes in recognizability should primarily be driven by children's experience drawing specific categories and in turn that frequently drawn categories should show the strongest developmental trends. To test this possibility, we first asked parents to report how often their child produces drawings of each category (*N* = 50 parents of children aged 3–10 years, see *Methods*, Supplementary Fig. 3). We did not find evidence that drawings of more frequently practiced categories were more

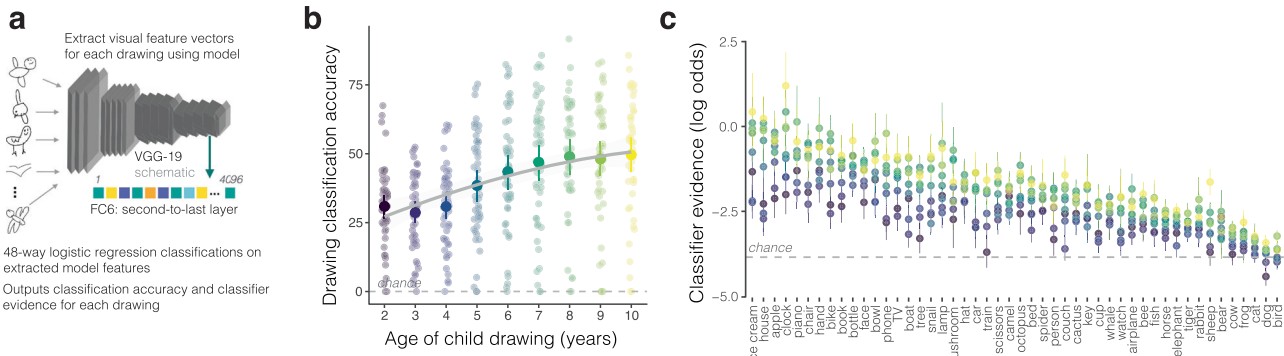

**Fig. 2 | Analyzing drawing accuracy over development. a** Overview of the analysis strategy for classifying children's drawings on the basis of VGG model embeddings. **b** Mean proportion of drawings recognized as a function of children's age; each dot represents the proportion of drawings that were correctly classified in a given category; the gray chance line represents 1/48 (number of categories in the dataset). Error bars represent 95% confidence intervals bootstrapped across the $N = 48$ categories for a total of $N = 22272$ drawings in the balanced dataset. **c** The y-axis represents the mean log-odds probabilities (i.e., classifier evidence)

assigned to the target category, binned by the age of the child (in years) who produced the drawing. Error bars represent 95% confidence bootstrapped across the number of drawings included in the balanced dataset for each age and category (average number of drawings in each bin $N = 51.6$, range = 14–102 drawings). Colors correspond to the ages shown on the x-axis of **b**. Categories on the x-axis are ordered by average log-odds probabilities for each category in descending order to highlight variation. A generalized mixed-effect model fit to recognition data from each drawing was used to analyze these data; all statistics are detailed in Table 1.

### Table 1 | Modeling children's visual production behavior

|  | Estimate | Std. Error | z value | Pr(>|z|) | 2.5% CI | 97.5% CI |
|---|---|---|---|---|---|---|
| Intercept | −0.690 | 0.173 | −3.979 | <0.0001 | −1.030 | −0.350 |
| Age (in years) | 0.251 | 0.020 | 12.805 | <0.0001 | 0.212 | 0.289 |
| Est. drawing frequency | −0.062 | 0.173 | −0.356 | 0.721 | −0.402 | 0.278 |
| Avg. tracing rating | 0.267 | 0.020 | 13.529 | <0.0001 | 0.229 | 0.306 |
| Time spent drawing | 0.039 | 0.021 | 1.868 | 0.062 | −0.002 | 0.079 |
| Ink used | −0.031 | 0.020 | −1.546 | 0.122 | −0.070 | 0.008 |
| Number of strokes | 0.008 | 0.018 | 0.477 | 0.634 | −0.026 | 0.043 |
| Age × Drawing frequency | 0.017 | 0.017 | 1.042 | 0.298 | −0.015 | 0.050 |

Results of a generalized linear mixed-effect model predicting the recognizability of each drawing (i.e., binary classification scores) from VGG-19 FC6 classifications, including random intercepts for each category and participant. All predictors were z-scored prior to analysis such that coefficients are standardized and comparable. All significance tests are Wald significance tests based on the coefficient values; these tests are two-tailed. No adjustments were made for multiple comparisons. See *Methods* for further model specifications.

recognizable or associated with stronger developmental trends (no fixed effect of drawing frequency or interaction with age in a generalized linear mixed-effects model, Table 1). This pattern of results was robust to the choice of model (Supplementary Table 3) and was replicated using human recognition scores in the controlled experiment (Supplementary Table 4). Instead, many infrequently drawn categories (e.g., ice cream) were well-recognized, while some frequently drawn categories (e.g., dog) were more likely to be confused with similar categories (see Fig. 2c).

Fig. 2c shows these developmental trends broken down by each category, highlighting wide variability across categories (Supplementary Fig. 4 for validation using CLIP); some categories were not well-recognized at all, especially in younger children (Fig. 2c). We next examined whether other measures of frequency of experience in children's daily life might predict this variation—for example, frequency in child-directed speech or all English-language books (Supplementary Table 5). However, we did not find a statistically significant relationship between these measures of frequency and the recognizability of children's drawings.

**Visuomotor control explains some changes in recognizability**
We anticipated that the recognizability of children's drawings would vary with children's ability to control and plan their motor movements. Children spend countless hours both learning to write and practicing how to produce different shapes. Children's engagement could also

reasonably vary as a function of age, with more skilled children spending more time, ink, or strokes on their drawings. We therefore measured the amount of effort children put into their drawings, and estimated children's visuomotor control via the shape tracing task. Children traced both a relatively easy shape (a square) as well as a complex, novel shape that contained both curved and sharp segments (see Fig. 1b). We then used each participant's tracings to derive estimates of their visuomotor control. To do so, we obtained ratings of tracing accuracy from adults for a subset of tracings and then used these ratings to adapt an image registration algorithm[53] to predict tracing scores for held-out tracings produced by children (see *Methods*). Tracing scores produced by the same participant were moderately correlated ($r(6754) = 0.60$, $t = 61.93$, $P < 0.001$, 95% CI = [0.586, 0.617]), despite the irregular shape being harder to trace than the square and the brevity of this tracing assessment.

If age-related changes in drawing recognizability primarily reflect changes in visuomotor control[35], then accounting for visuomotor control should explain away the age-related variance we observed. However, we still found a fixed effect of age after accounting for tracing abilities and effort covariates (Table 1), including the amount of time children spent drawing, the number of strokes in each drawing, and the amount of "ink" children used (see *Methods*); this effect was robust to model choice (validation using CLIP with fixed effect of age in Supplementary Table 3, examples in Supplementary Fig. 7, multicolinearity analysis in Supplementary Table 6). Even though children's

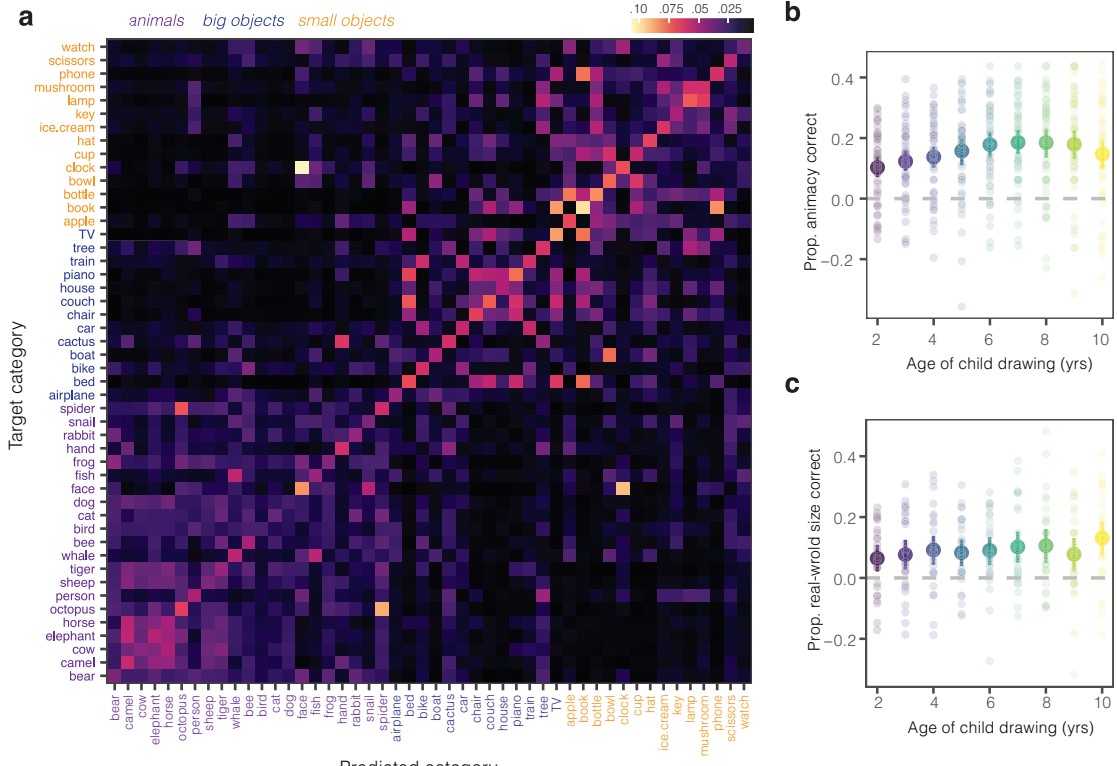

**Fig. 3 | Semantic information in misclassified drawings. a** Classifier probabilities for the subset of drawings that were misclassified on the basis of VGG-19 embeddings (FC6) ($N = 13682$ misclassified drawings produced by $N = 6270$ children, with 48 probabilities for each drawing). The y-axis shows the category children were intending to draw; the x-axis shows all of the categories in the dataset. Lighter values represent greater classifier probabilities assigned to a given category (see colorbar). **b**, **c** Proportion of misclassified drawings that contained the correct animacy/object size information of the target category (relative to baseline in the dataset); each dot represents the proportion of drawings in a given category that had correct animacy/real-world size information relative to baseline at each age, respectively. Error bars represent bootstrapped 95% confidence intervals across **b** $N = 48$ categories at each age ($N = 13682$ drawings total), and **c** across $N = 26$ inanimate categories at each age ($N = 6772$ drawings total).

ability to control and plan their motor movements predicts their ability to produce recognizable drawings, this factor does not fully account for the observed developmental changes.

## Recognizable drawings become more informative across development

The above results suggest that children gradually improve their ability to include diagnostic visual information in their drawings. However, they are also consistent with an account where younger children are just as able to produce recognizable drawings when they are engaged with the task, but are less likely to stay on task and thus produce unrecognizable drawings more often. To tease these two possibilities apart, we compared how much diagnostic visual information was contained in drawings that were correctly recognized.

For example, among drawings that were correctly recognized as clocks, did older children also include visual information that more clearly set them apart from other similar categories—for example, watches? We found that for correctly classified drawings (38.6% of the balanced subset of drawings, $N = 8590$), the amount of diagnostic information they contained still increased as a function of age, as measured by the log-odds probability assigned by the logistic-regression classifier to the target category ($\beta = 0.111$, SE = 0.015, df = 3544.18, $t = 7.354$, $P < 0.001$, 95% CI = [0.082, 0.141]; fixed effect of age in Supplementary Table 7, validation using CLIP in Supplementary Table 8). Age-related improvements in children's abilities to produce recognizable drawings also reflect a gradual increase in the amount of category-diagnostic information in their drawings.

## Misrecognized drawings still contain semantic information

Even if a child does not know the diagnostic features of giraffes or rabbits, they likely know that both are animals with four legs. This kind of coarse semantic information may still be contained in children's unrecognizable drawings. Indeed, prior work suggests that basic-level recognition—for example, recognizing something as a piano—is not a pre-requisite for inferring semantic information about a depicted object, such as whether it refers to something alive or its size in the real-world. Adults can reliably judge the animacy and real-world size of unrecognizable, textured images by inferring that animals tend to have high curvature and that larger, inanimate objects tend to have boxier shape structures[54,55] and preschool children appear sensitive to these cues[56].

Following this idea, we reasoned that even young children's misclassified drawings might contain information about the animacy and real-world size of the category they tried to draw (see Fig. 3a). We thus analyzed the patterns of drawing misclassifications, examining the animacy and real-world size of the incorrect category that was assigned the highest probability for each drawing. We found that misclassified drawings reliably carried information about the animacy and real-world size of the category children were trying to draw (see Fig. 3b, c). Both animacy and real-world object size information was decodable across all ages (all 95% CIs shown in 3b, c do not include chance) though we found wide variation across individual categories (each data point in Fig. 3b, c). More broadly, we observed structure in the pattern of probabilities assigned by the classifier to the other categories (see Fig. 3a): for example, unrecognizable drawings of an octopus were often assigned a high classifier probability for a spider.

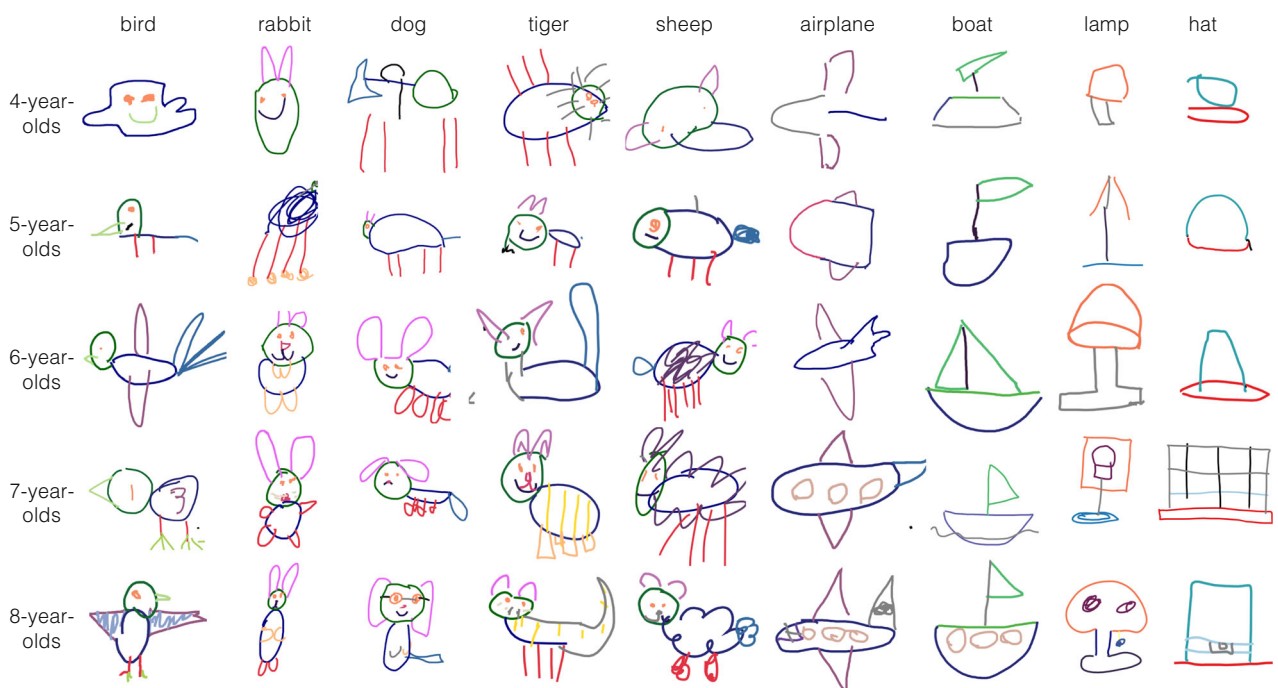

**Fig. 4 | Annotated drawings.** Example drawings from 4–8 year-old children, with part annotations. Each color represents object parts labels agreed upon by human annotators; gray lines represent strokes with multiple parts, and black lines represent unintelligible strokes.

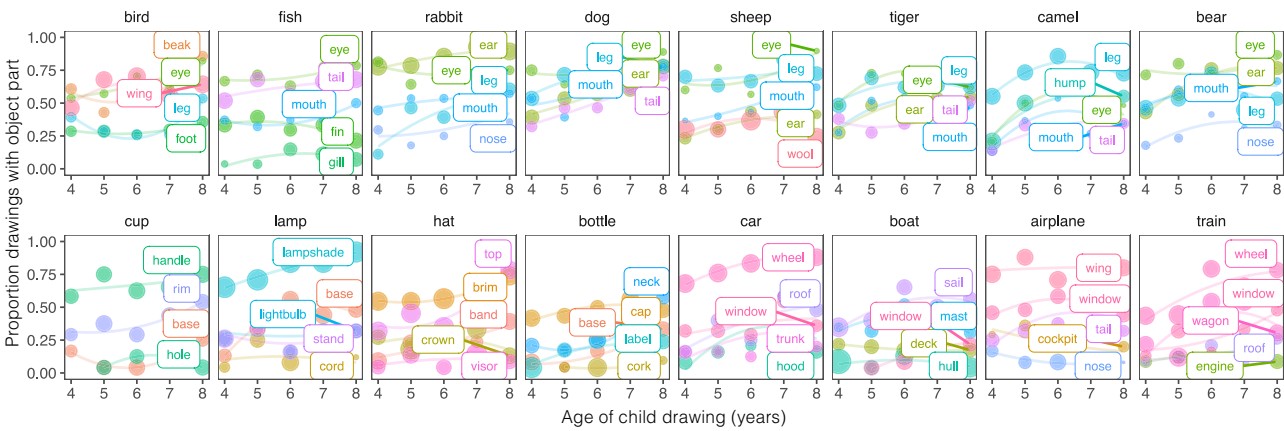

**Fig. 5 | Changes in object part inclusion and emphasis over development.** Proportion of drawings that included a given object part for each object category as a function of children's age (see *Methods*), analyzing annotations from $N = 1824$ drawings where annotators agreed on part labels (see *Methods*). The size of each dot reflects the average emphasis (proportion of stroke length relative to the entire drawing) for each object part within each bin (max plotted part emphasis = 0.5); the top five most frequent object parts are included for each category excluding generic "body/head" parts.

## Drawings contain more recognizable parts across development

What aspects of these drawings account for this improvement? A natural possibility is that children gradually learn which object parts to include and how much to emphasize those parts (e.g., long ears for rabbits) in their drawings[57]. We thus collected annotations of the visible object parts in a subset of $N = 2021$ drawings, and examined developmental changes in which parts children prioritized in their drawings throughout development (Fig. 4).

We found that drawings produced by older vs. younger children generally contained more semantic parts (fixed effect of age in a linear mixed-effect model, $\beta = 0.395$, SE = 0.041, df = 2071, $P < 0.001$, CI = [0.314, 0.476]); (Fig. 5, Supplementary Fig. 8). For some concepts, these gains appeared to be specific to single part:

for example, older children were more likely to produce cups with handles and cars with recognizable wheels. For other concepts, however, age-related changes were more complex: while most younger children's drawings of rabbits included recognizable ears, many of them were still not recognizable as rabbits. Thus while we observed clear age-related changes in the part complexity of children's drawings, the mere presence of—or amount of emphasis on— any particular part may not be sufficient to account for developmental variation in recognizability (see Supplementary Fig. 9); instead, children are likely also learning how to arrange several object parts to convey a recognizable exemplar. For example, the ears on rabbits may need to be more elongated relative to the head to provide a strong enough cue to category membership.

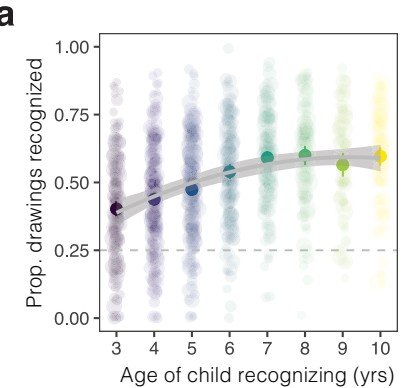
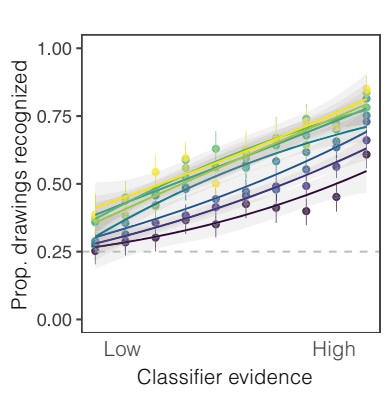
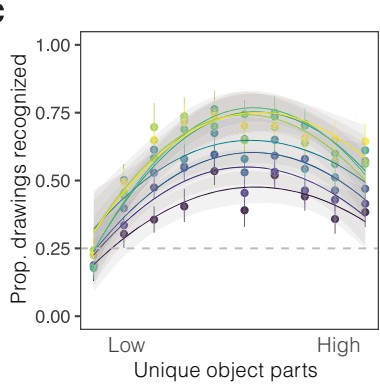

**Fig. 6 | Developmental changes in drawing recognition. a** Drawing recognition as a function of the age of the child who participated in the guessing game; each dot represents data from one child who participated and is scaled by the number of trials they completed. Error bars overlaid on top of the data represent 95% confidence intervals bootstrapped across the visualized raw data from individual participants in each age bin, with $N = 1789$ participants over all ages ($N = 36,615$ trials total). **b, c** Drawing recognition data plotted separately by the age of the child participating as a function of the **b** amount of diagnostic visual information in each drawing, operationalized as the classifier evidence assigned to each sketch and **c** the number of unique object parts in each drawing. Both variables are binned into deciles for visualization purposes. Error bars represent 95% confidence intervals bootstrapped over the recognition performance for all drawings included in each combination of bin/age group. Colors of each line correspond to the ages shown on the x-axis of **a**.

## Drawing recognition performance improves across development

Why do children include more diagnostic visual information in their drawings as they grow older? One source of these changes may be refinements in children's internal visual concepts. Children might come to more clearly represent the visual information that best distinguishes depictions of rabbits from dogs, for example, and may be able to use this information when recognizing drawings. If so, children should improve over development in their ability to exploit visual information in drawings to recognize their intended meaning.

To test this idea, we installed a "guessing game" in the same kiosk (see Fig. 1c) where children guessed the category that an earlier child's drawing referred to. These drawings were randomly sampled from the larger drawing dataset and thus varied in the amount of diagnostic visual information they contained. Our goal in designing this task was for it to be challenging yet not demand that children track a large number of comparisons. At the beginning of each session, children completed four practice trials in which they were cued with a photograph and asked to "tap the [vehicle/animal/object] that goes with the picture," choosing from an array of four photographs of different visual concepts (see Fig. 1c). Children were then cued with drawings of these categories and responded using the same photograph buttons; photograph matching trials were also interspersed throughout as attention checks. We sequentially deployed four different versions with different sets of four perceptually similar categories in each (see *Methods*). We restricted our analyses to data from children age 3 and above, having observed that children under 3 were less likely to engage with the task independently. After exclusions, the resulting dataset included 1,789 children ages 3–10 years-old (see *Methods*).

We found that children steadily improved across development at identifying the category that a drawing referred to (see Fig. 6a), though our youngest participants (3-year-olds) performed only slightly higher than chance (25%). In contrast, performance on photograph matching catch trials was relatively similar across ages. All children whose data were included in our analyses scored greater than 75% correct on photograph trials and average accuracy in each group ranged from $M = 90$–93% correct. Thus, variation in drawing recognition accuracy is unlikely to be explained by generic differences in motivation or task engagement.

Does children's ability to exploit category-diagnostic visual information during recognition improve over childhood? We examined how children's drawing recognition abilities varied with respect to the amount of diagnostic visual information in a given drawing. For each drawing that appeared in the guessing games, we measured diagnostic visual information via a 4-way logistic regression classifier trained on the VGG-19 features extracted from the drawings in each guessing game (see *Methods*). That is, the diagnostic information for a dog drawing was defined relative to its perceptual similarity to the other choices in the recognition task (i.e., bird, fish, rabbit). We then fit a generalized linear mixed effects models predicting children's recognition performance with child's age, this metric of diagnostic visual information, and their interaction as fixed effects (see *Methods*).

Drawings with more diagnostic visual information were better recognized across all ages (fixed effect of classifier evidence in Table 2, Fig. 6b, CLIP robustness check in Supplementary Fig. 10, Supplementary Table 9, see also Supplementary Fig. 11). Yet older children were also better able to capitalize on graded differences in the diagnostic visual information in drawings when recognizing them (see Fig. 6b; interaction between fixed effects of classifier evidence and recognizer age in Table 2). This result held when we restricted our analyses to children who performed at ceiling on photograph matching trials (interaction between age and classifier evidence in Supplementary Table 10) suggesting that these effects are unlikely to be driven by a

### Table 2 | Modeling children's visual recognition behavior

|  | Estimate | SE | z value | Pr(>\|z\|) | 2.5% CI | 97.5% CI |
|---|---|---|---|---|---|---|
| Intercept | 0.050 | 0.121 | 0.412 | 0.68 | −0.187 | 0.287 |
| Classifier evidence | 0.477 | 0.046 | 10.405 | <0.0001 | 0.387 | 0.566 |
| Recognizer age | 0.317 | 0.019 | 16.777 | <0.0001 | 0.280 | 0.354 |
| Classifier evidence × Recognizer age | 0.062 | 0.014 | 4.246 | <0.0001 | 0.033 | 0.090 |

Model coefficients of a generalized linear mixed-effects model predicting binary visual recognition performance for each drawing as a function of recognizer age and classifier evidence in each drawing that was recognized by children. All predictors were z-scored prior to analysis such that coefficients are standardized and comparable. All significance tests are Wald significance tests based on the coefficient values; these tests are two-tailed. No adjustments were made for multiple comparisons. See *Methods* for further model specifications.

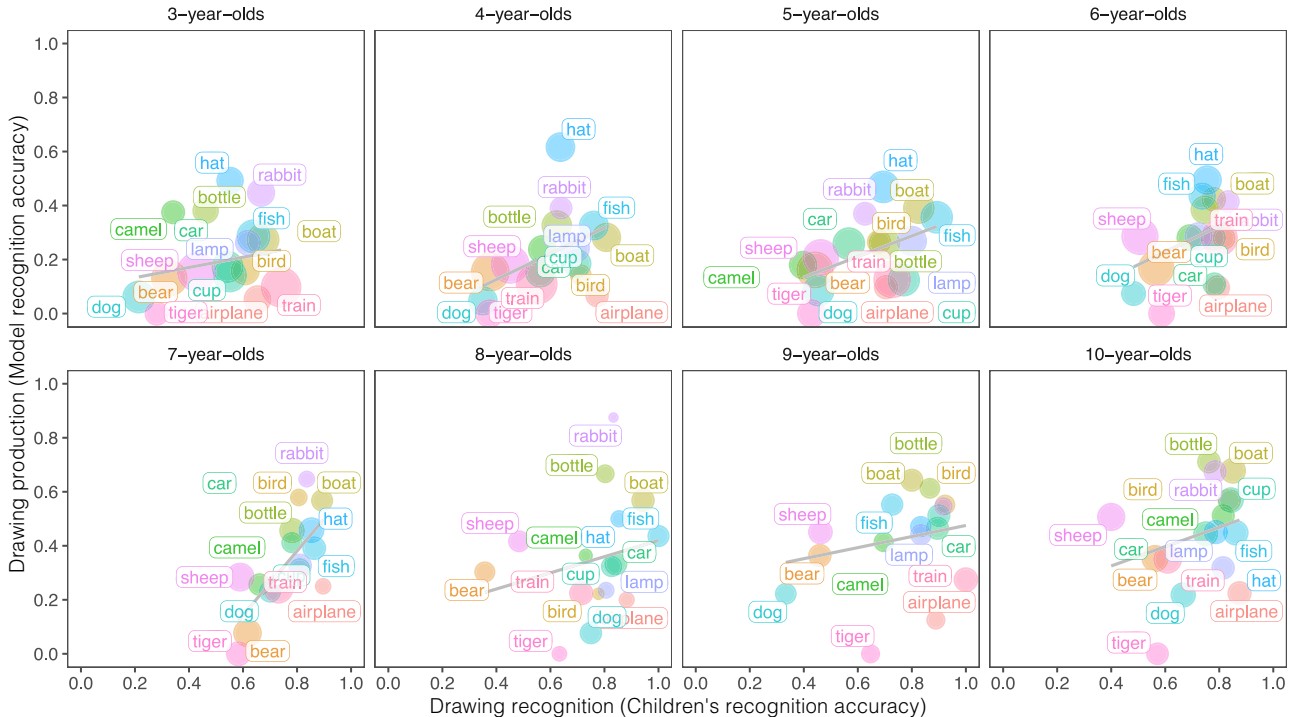

**Fig. 7 | Relating drawing production and recognition.** Each dot represents a category (e.g., hat) at a given age (in years), where the y-axis value represents how well children of that age produced recognizable drawings of that category (as assessed by CLIP model classifications, $N = 8674$ drawings) and the x-axis value represents how well children of that age were able to recognize the top 30% most recognizable drawings of that category (as assessed by accuracy in the 4AFC recognition games, $N = 3221$ drawings). The size of each dot represents the number of drawings included per category in model recognition accuracy. Independent sets of drawings are analyzed in each case.

differences in either engagement or in the ability to match drawings with the picture-cue buttons (Supplementary Fig. 12).

Does children's ability to use object part information during visual recognition also change across development[15,24]? We examined how drawing recognition accuracy varied with the number of unique object parts included in the drawings children tried to recognize. We again fit a generalized linear mixed-effect model to the recognition data, modeling the interaction between the number of unique parts in each drawing and the age of the child recognizing the drawing.

Drawings with more unique object parts tended to be better recognized—but, unexpectedly, drawings with many object parts were harder to recognize than drawings with an intermediate number of parts (Fig. 6c), though there was substantial variation across categories (Supplementary Fig. 13, Supplementary Table 11). Critically, we again found that older children were better able to capitalize on increasing object part information (interaction between number of unique parts and recognizer age in Supplementary Table 11). Children's ability to integrate additional object part information during recognition changed across development.

### Relationship between visual production and recognition

To what degree do changes in children's production and recognition of drawings reflect changes in the same mental representations? Insofar as children's abilities to recognize what drawings mean and to produce meaningful drawings both rely on a shared representation, then their ability to produce a drawing of a dog may be correlated with their ability to recognize a drawing of a dog, as in adults[41]. To explore this idea, we borrow an analysis technique used in language acquisition[58], where variation in word production is well-predicted by independent data about that word.

We thus explored how well variation in visual production is related to visual recognition at the category-level, acknowledging the exploratory nature of these analyses. To do so, we examined children's

visual production and recognition abilities using independent sets of drawings of the same categories. Note that while some children may have contributed drawings and participated in the recognition games (with different categories), these sessions were anonymous and we do not have access to within-child data. To estimate drawing recognition abilities, we used children's performance on the guessing games to calculate how often children of a given age, on average, were able to recognize drawings of a given category. To ensure that we were examining recognition for relatively recognizable drawings, we analyzed how well children could recognize the top 30 percent most recognizable drawings of each category (see *Methods*).

Visual production and visual recognition abilities were positively related at the category-level (when aggregating across age, $r = 0.53$, $t(14) = 2.362$, $P = 0.033$, CI $= [0.05, 0.81]$, see results broken down by age in Fig. 7). For example, while dogs and sheep were both harder to produce and to recognize, rabbits and hats were easier to produce and recognize. We thus find relative consistency across categories in these two tasks, suggesting that children's ability to perform well in both tasks may rely on a shared visual representation, and paving the way for future work that seeks to understand the sources of this category variation using within-child, controlled experiments.

## Discussion

We conducted a systematic investigation of how children produce and recognize line drawings of a wide range of visual concepts across development. We developed a dataset of children's drawings (>37K) and capitalized on innovations in machine learning to quantify changes in children's drawings across development. We found robust improvements in children's ability to include diagnostic visual information via recognizable object parts in their drawings, and these developmental changes were not reducible to either increased effort or better visuomotor abilities. Further, we found that children's unrecognizable drawings contained information about the animacy

and real-world size of the visual concepts they were trying to depict, highlighting an intermediate stage between scribbles and recognizable drawings. We also found improvements throughout childhood in children's ability to recognize each other's drawings, particularly in their ability to capitalize on diagnostic visual information during drawing recognition. Together, these results document parallel developmental changes in how children use diagnostic visual information when producing and recognizing freehand line drawings, suggesting that refinements in children's visual concepts may underlie improvements across both tasks.

More broadly, the present work highlights how the combination of modern machine learning methods and larger-scale datasets of naturalistic behaviors can contribute to theoretical progress in developmental science. By collecting rich data from many participants over a large developmental age range, we can more precisely estimate graded changes in children's abilities and the degree to which these trajectories vary across categories. In turn, our use of innovations in computer vision and computational modeling allow the analysis of the entirety of this large dataset, capturing variation across both unrecognizable and recognizable drawings in a single analytic approach (which would have been intractable with human ratings). Using this approach, we were able to distinguish variability in children's drawings due to a range of different developmental processes—including motor skill and task effort—from variability related to visual concept knowledge. We believe that this work paints a more accurate picture of developmental change and opens up new avenues for investigating the various factors that shape visual concepts throughout development both using large-scale datasets and controlled, within-child experiments that directly relate visual production and recognition and examine item variation.

Our exploratory analyses suggested that children's abilities to produce and recognize drawings were correlated at the category-level, e.g., drawings of dogs were both harder to produce and to recognize. Further, estimates of how often these items tend to be drawn or experienced did not explain this variation. Why might some categories be easier to draw and to recognize?

First, these item effects may be related to other metrics of experience beyond what we have measured: Perhaps exposure frequency in educational materials or children's media were not adequately captured in our analyses. Or perhaps children may experience more invariant exemplars of some categories, making it easier to identify and draw those categories. For example, while dogs may vary substantially, ice cream cones have a relatively more invariant form. Children may develop more refined visual representations for more frequently experienced and more invariant items, leading to more recognizable drawings.

Any effects of exposure frequency or form variability might interact with the degree to which a category has a 3D shape structure that can be easily depicted using a line drawing. For example, canonical mushrooms have a relatively simple shape, whereas rabbits have many sub-parts that need to be accurately depicted and arranged (i.e., head, ears, nose). In turn, these shape structures may lead different categories to have more iconic representations that children access when producing and recognizing drawings. For example, trains are often depicted as steam trains, as modern trains can be hard to distinguish from other vehicles as line drawings.

The present work also highlights the gradual progression in children's drawings from exploratory scribbles through an intermediate stage where their drawings may not unequivocally convey a specific visual concept (e.g., a giraffe), while still containing enough visual information to be recognizable as an "animal." Drawing tasks may allow children to convey this kind of partial knowledge about a visual concept that may otherwise be difficult for them to express verbally. And as children learn more about a specific visual concept—for example, that giraffes have longer legs than antelopes—these gains in

conceptual knowledge may manifest in their drawings. Drawing production tasks may thus hold potential for examining graded changes in children's visual concept knowledge over development.

Several learning mechanisms are consistent with the developmental changes we observed. First, children likely become better visual communicators as they learn which visual features are most effective at conveying category membership through the process of producing drawings. Children may also become more aware that drawings can serve a communicative function and may start primarily drawing for others versus for themselves. In turn, this process of using drawings to communicate may have downstream effects on children's ability to recognize drawings. Indeed, both drawing experts[43] and naïve adults who practice drawing similar categories[39] show enhanced visual recognition abilities of these categories. Such a mechanism would be consistent with prior work suggesting that learning to produce letters by hand can support subsequent letter recognition[59,60]. Contra a strong version of this account, however, we did not find effects of drawing practice at the category level in the present data: for example, ice cream cones were among the best-recognized categories and estimated (by parents) to be among the least practiced by children.

A second, non-exclusive possibility is that children are explicitly learning the diagnostic features of categories as they enrich their semantic knowledge. For example, children may learn about the functional properties of different attributes: camels have humps to store water, and clocks have numbers to tell time. In turn, these changes in semantic knowledge could percolate into children's visual concepts and be accessed both when children draw an object and when they recognize it. Indeed, children change in how they think about the diagnosticity of different semantic properties across development: for example, in early childhood, the fastest cheetah tends to be seen as the best and the most representative cheetah[61]. Taxonomic groupings also become increasingly important in children's explicit and implicit conceptual judgements[12,62]. Children's evolving semantic knowledge could thus shape the visual features children use both when producing and recognizing different visual concepts.

A third possibility, again not mutually exclusive with the other two, is that children are implicitly learning category-diagnostic information through repetitively viewing and categorizing depictions, real-life examples, and photographs of these different categories. Indeed, the neural networks used here did not have visuomotor experience drawing or training about the semantic properties of these categories. Thus in principle it is possible that children could be refining their visual concepts without substantial involvement from other cognitive or sensorimotor systems.

There are various limitations to the generalizability of these findings that future work could address. First, while these datasets are large and sample heterogeneous populations, all drawings and recognition behaviors were collected at a single geographical location, limiting the generalizability of these results to children from other cultural or socioeconomic backgrounds[63]. Children in different contexts may spend considerably more or less time viewing and producing depictions of different categories, and different cultural contexts have different conventions for depicting visual concepts[64]. Yet some aspects of drawing production and interpretation are likely shared across cultural contexts[4,65], given prior work that has investigated picture comprehension in communities with modest exposure to Western visual media[66,67]. Moreover, there is evidence from earlier work that some of this convergence may reflect evolutionarily conserved visual processing mechanisms, as non-human primates can recognize the correspondence between line drawings and their real-world referents[68]. Future work that examines drawings across different cultural contexts in both adults[64] and children will help quantify the consistency and variability in how we represent and depict visual concepts.

Second, while we imposed strong filtering requirements, we were not present while the children were drawing or guessing at the kiosk

and thus cannot be sure that we eliminated all sources of noise or interference. Many sources of additional interference would only generate noise in our data, though, rather than creating specific age-related trends. Nonetheless, we replicated our main experimental results on drawing production in a controlled, experimental context with a smaller set of categories (see Supplementary Fig. 6).

Third, since these datasets are cross-sectional, they do not directly relate visual production and recognition abilities at the individual level. Our exploratory, category-level analyses suggest variation in these two abilities are correlated across development; ultimately, however, within-child measurements will be necessary to confirm that changes in children's visual concepts underlie the observed changes in both tasks. In addition, these correlational analyses can only provide hints as to whether changes in visual production cause changes in visual recognition or vice versa. Finer-grain, within-child training studies (as in ref. 30) could provide traction on the direction of causality between visual production and recognition.

Overall, our results call for further systematic, experimental investigations into the kinds of experience–including visuomotor practice, semantic enrichment, and visual exposure–that may influence visual production and recognition in children, and we hope that the open datasets and tools we have created here will open up new avenues for such future work. We propose that a full understanding of how children produce and recognize drawings of visual concepts will allow a unique and novel perspective on the both the development and the nature of visual concepts: the representations that allow us to easily derive meaning from what we see.

## Methods

### Ethics approval
All research presented here was approved by the Institutional Review Board at Stanford University under Protocol 43992, Development of Children's Drawing Abilities.

### Drawing station details
While the interface was designed to be navigable by children, the first page of the drawing station showed a short waiver of consent form that parents completed and asked parents to enter their child's age in years; no other demographic information was collected. Afterwards, video prompts of an experimenter guided the child through the rest of the experiment; an initial video stated that this game was "only for one person at a time" and asked children to "draw by themselves." Every session at the drawing station started with tracing trials before moving on to the category prompts ("What about a [*couch*]? Can you draw a [*couch*]?"). Children could stop the experiment at any time by pressing a stop button; each trial ended after 30 seconds or after the child pressed the "next" button. Six different sets of eight category prompts rotated at the station, yielding drawings from a total of 48 categories (see Supplementary Fig. 1, airplane, apple, bear, bed, bee, bike, bird, boat, book, bottle, bowl, cactus, camel, car, cat, chair, clock, couch, cow, cup, dog, elephant, face, fish, frog, hand, hat, horse, house, ice cream, key, lamp, mushroom, octopus, person, phone, piano, rabbit, scissors, sheep, snail, spider, tiger, train, tree, TV, watch, whale); these categories were also chosen to overlap with those in the QuickDraw database of adult drawings     (https://github.com/googlecreativelab/quickdraw-dataset). Each set of category prompts that rotated at the station thus included both animate and inanimate categories as well as commonly and infrequently drawn categories; category prompts were presented in a random order. No statistical method was used to predetermine sample size; rather, we aimed to gather as large of a dataset as possible and data collection was stopped in March 2020 due to COVID-19.

### Drawing dataset filtering & descriptives
Given that we could not easily monitor all environmental variables at the drawing station that could impact task engagement (e.g., ambient noise,

distraction from other museum visitors), we anticipated the need to develop robust and consistent procedures for data quality assurance. We thus adopted strict screening procedures to ensure that any age-related trends we observed were not due to differences in task compliance across age. Early on, we noticed an unusual degree of sophistication in 2-year-old participants' drawings and suspected that adult caregivers accompanying these children may not have complied with task instructions to let children draw on their own. Thus, in subsequent versions of the drawing game, we surveyed participants to find out whether another child or an adult had also drawn during the session; all drawings where interference was reported were excluded from analyses.

Out of 11797 subsequent sessions at the station, 719 reported interference from either parents or other children; see Supplementary Table 2 for a detailed breakdown of reported interference by each type for each age group. These participants' drawings were not rendered or included in analysis. When observing participants interacting with the drawing station, we noted that most children's parents did not fill out the survey because they were either talking to other parents or taking care of a sibling. Further, while children could contribute drawing data more than once if they chose, this did not occur during our structured observation of the kiosk.

Raw drawing data were then screened for task compliance using a combination of manual and automated procedures (i.e., excluding blank drawings, pure scribbles, and drawings containing words). A first subset of drawings ($N = 15,594$ drawings) was filtered manually by one of the authors, resulting in $N = 13,119$ drawings after exclusions (15.8% exclusion rate); subsequently, drawing filtering was crowd-sourced via Prolific. 390 participants first completed a practice round demonstrating valid and invalid drawings and then viewed 24 drawings from a intended category at a time and selected the invalid drawings they judged to come from from off-task participants. Participants were reminded that unrecognizable drawings were still "valid" drawings, and could proceed to the next category only after selecting a "catch" invalid drawing. Each drawing in the dataset was viewed at least twice by two different participants. To be conservative, any drawing that was marked as 'invalid' by a participant was excluded from the dataset. These stringent filtering criteria resulted in the exclusion of an additional 9897 drawings, leading to an overall exclusion rate of 24.57% of the drawings and a final set of 37,770 drawings from 8084 sessions. In the final dataset, there were more younger than older children, despite filtering; see Supplementary Table 1 for a complete summary.

### Experimental dataset procedure
In a separate experiment[52,69], children were seated in front of a touchscreen tablet with a trained experimenter. As in the larger dataset, children completed two shape-tracing trials, and then children produced drawings of 12 familiar object categories (airplane, bike, bird, car, cat, chair, cup, hat, house, rabbit, tree, watch) which were randomly assigned to different cue-types (verbal vs. picture). In this paper, we analyze only verbal-cued drawings for sake of comparison to the drawing station dataset. 135 children participated in the experiment; 6 participants were excluded, including 3 for skipping more than 6 drawing trials and 3 for scribbling three or more times in a row. Six additional participants were tested but their data was not recorded due to a technical error, and two participants never advanced past the practice trials, leading to a final sample of 121 children. Approximately twenty participants were included in each age group (i.e., twenty 4-, 5-, 6-, 7-, 8-, and 9-year-olds); see full breakdown in ref. 52. No additional demographic data was recorded about the participants. This protocol was also approved by the Institutional Review Board at Stanford University (43992, Development of Children's Drawing Abilities).

### Measuring tracing accuracy
We developed an automated procedure for evaluating how accurately participants performed the tracing task that was validated against

empirical judgments of tracing quality. We decompose tracing accuracy into two components: a *shape error* component and a *spatial error* component. Shape error reflects the degree to which the child's tracing matched the contours of the target shape; the spatial error reflects the degree to which the location, size, and orientation of the child's tracing matched the target shape.

To compute these error components, we applied an image registration algorithm, AirLab[53], to align each tracing to the target shape. This yielded an affine transformation matrix that minimized the pixel-wise correlation distance between the aligned tracing, $T$, and the target shape, $S$: $Loss_{NCC} = -\frac{\sum S \cdot T - \sum E(S)E(T)}{N \sum Var(S)Var(T)}$, where $N$ is the number of pixels in both images.

The shape error is then the final correlation distance between the aligned tracing and the target shape. The spatial error is then the magnitude of three distinct error terms: location, orientation, and size error, derived by decomposing the affine transformation matrix above into translation, rotation, and scaling components, respectively. This procedure generated four error values for each tracing: one value representing the shape error (i.e., the pixel-wise correlation distance) and three values representing the spatial error (i.e., magnitude of translation, rotation, and scaling components).

Although we assumed that both shape and spatial error terms should contribute to our measure of tracing task performance, we did not know how much weight to assign to each component to best predict empirical judgments of tracing quality. In order to estimate these weights, we collected quality ratings from adult observers ($N = 70$) for 1325 tracings (i.e., 50–80 tracings per shape per age), each of which was rated 1–5 times. Raters were instructed to evaluate "how well the tracing matches the target shape and is aligned to the position of the target shape" on a 5-point scale.

We fit an ordinal regression mixed-effects model to predict these 5-point ratings, which contained correlation distance, translation, rotation, scaling, and shape identity (square vs. star) as predictors, with random intercepts for rater. This model yielded parameter estimates that could then be used to score each tracing in the dataset ($N = 14372$ tracings from 7612 children who completed at least one tracing trial). We averaged tracing scores for both shapes within each session to yield a single tracing score for each participant.

## Measuring effort covariates

For each drawing trial, children had up to 30 seconds to complete their drawings with their fingers. We recorded both the final drawings and the parameters of each stroke produced by children at the drawing station, allowing us to estimate the amount of time children put into their drawings. As a second measure of effort, we also counted the number of strokes that children put into a given drawing. Finally, we estimated the proportion of the drawing canvas that was filled (e.g., 'ink used') by computing the proportion of each final drawing that contained non-white pixels.

## VGG-19: visual encoder

To encode the high-level visual features of each sketch, we used the VGG-19 architecture[48], a deep convolutional neural network pretrained on Imagenet classification. For our main analysis, we used model activations in the second-to-last layer of this network, which is the first fully connected layer of the network (FC6), as prior work suggests that it contain more explicit representations of object identity than earlier layers[40]. Raw feature representations in this layer consist of flat 4096-dimensional vectors. to which we applied channel-wise normalization across all filtered drawings in the dataset. For additional analyses using the earlier convolutional layers, we first applied spatial averaging over the outputs of each layer to reduce their dimensionality, as in ref. [40], before also applying channel-wise normalization.

## VGG-19: logistic regression classifiers

Next, we used these features to train object category decoders. To avoid any bias due to imbalance in the distribution of drawings over categories (since groups of categories ran at the station for different times), we sampled such that there were an equal number of drawings of each of the 48 categories ($N=22,272$ drawings total). We then trained a 48-way logistic classifier with L2 regularization (tolerance = 0.1, regularization = 0.1), and used this classifier to estimate the category labels for a random held-out subset of 96 drawings (2 drawings from each category). No additional metadata about the age of the child who produced each sketch was provided to the decoder. This procedure was repeated for entire dataset ($K = 232$ fold) yielding both a binary a recognition score and the softmax probability assigned to each target class in the dataset. We define classifier evidence as the log-odds ratio of the probability assigned to the target category vs. the other categories in the dataset; this metric thus captures the degree to which a given drawing contains visual information that is diagnostic of the target category (and not of the other categories in the dataset); these log-transformed values are also more suitable for the linear mixed effects models used in analyses.

## CLIP classifications

CLIP classifications were obtained by assessing the similarity between model embeddings for each sketch to each category label, as in ref. [49]. This method thus also yields both binary classification scores and probability scores for each of the 48 categories in the dataset.

For these analyses, we used the ViT-B/32 implementation of CLIP publicly available at https://github.com/openai/CLIP. Model features were extracted for center-cropped versions of each sketch in the entire dataset ($N = 37,770$) and for the tokenized text versions of the labels for each of the 48 categories (e.g., "a dog"). We then computed the cosine similarity between the features for each sketch and each of the 48 category labels and assessed which category label received the highest similarity. If the category label that had the highest similarity was the category children were prompted to draw, this was counted as a correct classification.

## Human recognition scores: experimental dataset

We measured the recognizability of each drawing in the controlled, experimental dataset via an online recognition experiment. Adult participants based in the U.S. were recruited via Prolific for a 15-min experiment and asked to identify the category depicted in a random subset of approximately 140 drawings; each drawing was shown to 10 participants. No statistical method was used to predetermine sample size. Participants were shown these drawings in a random sequence and asked "What does this look like?" and selected their responses from the set of 12 categories and were encouraged to provide their best guess if they were unsure. No participants were excluded from analysis for missing the catch trial, which was included to verify that participants could accurately describe their goal in this task. We then computed a recognition score for each drawing, reflecting the proportion of participants who correctly identified the target category.

## Mixed-effect models

Two mixed effects models were fit to assess the degree to which children produced more recognizable drawings across childhood. A first generalized mixed-effect model was fit to the binary classification scores for each drawing using a logit linking function. A second linear mixed-effect model was fit to the log-odds target probability assigned to each drawing, restricting our analyses to correctly classified drawings. In both cases, we included fixed effects of children's age (in years), estimated drawing frequency for each category (via parental report), their interaction, children's estimated tracing score (see above), the time children spent drawing (in seconds), the mean intensity of the drawing (i.e., percentage of non-white pixels), and the

number of strokes children used. All predictors were scaled to have a mean of 0 and a standard deviation of 1, making coefficients interpretable relative to each other. Random intercepts were included for each participant and each category.

Generalized linear mixed effects models were run using the *lme4* code packages[70] in R, where *P*-values are based on asymptotic Wald tests, which is standard practice for generalized linear models. Linear mixed-effect models were run using *lmerTest*, where p-values were estimated via Satterthwaite's degrees of freedom method. All significance tests were two-tailed; we did not correct for multiple comparisons. Residuals for each model were examined to ensure that the model was not biased, and predictors were examined to ensure that they were not colinear (see Supplementary Table 6). Full model specifications and code package versions are included in the repository at https://osf.io/qymjr/.

### Animacy & object size information in misclassified drawings
For each misclassified drawing, we calculated whether the category assigned by the logistic regression classifier was the same animacy as the target category, assigning a binary animacy classification score for each drawing. The same procedure was repeated for inanimate objects with respect to their real-world size (big: approximately larger than a chair, small: could be held with one hand)[54]. These binary scores were averaged for each age and category, yielding a value between 0 and 1 representing the proportion of the drawings that were identified as having the correct animacy/size. As the proportion of animals/inanimate objects and big/small inanimate objects was not exactly balanced in the dataset, we subtracted the baseline prevalence for each broad category (i.e for animals, inanimate objects, big objects, and small objects) from this proportion. These values are plotted in Fig. 3b, c, as are the bootstrapped 95% confidence intervals calculated across all 48 categories using the baseline-corrected category values.

### Visual recognition task
On each trial of the guessing game, a photograph or drawing of an object category was presented on the screen, and children were asked to "tap the [animal/vehicle/object] that goes the with the [drawing/picture]"; response choices were indicated by circular buttons that contained photographs of canonical exemplars from each category, as well as the name of the category written above; the position of these response buttons was randomized for each participant. A fifth response choice was a button with a question-mark icon that could be used by participants to indicate they didn't know which category the drawing belonged to. To familiarize participants with the interface, the first four trials of every game were four photograph trials, one for each of the response choices. To encourage accurate guessing, a pleasant sound was played when the correct category was chosen, and the box surrounding the image briefly turned green; no feedback was given for incorrect trials. Every ten trials, a catch trial appeared where participants were required to match a very similar photograph to the photographic response buttons.

### Visual recognition task: drawing selection
We selected four subsets of categories for the guessing game at the station: small animals (dog, fish, rabbit, bird), vehicles (train, car, airplane, boat), small, inanimate objects (hat, bottle, cup, lamp), and large animals (camel, sheep, bear, tiger). Each version of the guessing game ran separately for approximately 2 months. For each game, we randomly selected drawings (20–25 per category, depending on availability) made by children ages 4-8 at the drawing station. We chose this age range to cover a wide range of drawing abilities and to ensure equal numbers of drawings were included per age group (as 9–10 year-old's are infrequent visitors to the museum). This resulted in 516−616 drawings for each guessing game from which 48 drawings were randomly sampled for each participant (8 drawings made by 4-,5-,6-,7-,

and 8-year-olds). If children completed the entire session, this resulted in a total of 48 trials for each participant (40 drawing trials and 8 photograph matching trials).

### Recognition data inclusion
As with the drawing data, we excluded any sessions where there was reported interference from parents or other children. As 2-year-old's showed significantly better performance than 3-year-old's in our first two guessing games—signaling some interference from their caregivers or siblings that was not reported in the surveys –we chose to exclude 2-year-old's from subsequent analyses. We excluded children who started the game but did not complete more than 1 trial after the practice trials ($N = 1068$ participants) and the 238 adults who participated. We also excluded all trials with reaction times slower than 10 s or faster than 100 ms, judging these to be off-task responses. Next, we excluded participants on the basis of their performance on practice and catch photograph matching trials. Given that these catch trials presented a very easy recognition task, we excluded participants who did not achieve at least 75% accuracy on these trials ($N = 795$). The remaining 1789 participants who met this criterion completed an average of $M = 21.69$ trials. On total, we analyzed 36,615 trials where children recognized each other's drawings. No statistical method was used to predetermine sample size. These analysis choices were pre-registered after examining data from two of the guessing games and then applied to the entire dataset (see registrations on https://osf.io/qymjr/).

### Recognition data analyses
To calculate the classifier evidence associated with each sketch that children recognized, we used the same visual encoder to extract visual features for each sketch (see *Visual Encoder*), and iteratively trained logistic regression classifiers (see *Logistic Regression Classifier*). For these analyses, we restricted the classification set to the drawings that were presented in each version of the guessing game to match the task conditions of the guessing game. We trained a separate logistic regression for each sketch that was presented using leave-one-out cross-validation. This procedure thus yielded probabilities assigned to each of four categories in each guessing game; these probabilities were used to calculate the log-odds ratios for the target category of each sketch which we refer to as classifier evidence. Due to random sampling, not every sketch included in the game had valid guesses associated with it; these sketches were thus not included in analyses. We then modeled children's recognition behavior in a generalized linear mixed-effect model, where recognizer age (in years), classifier evidence, and their interaction were specified as fixed effects. All predictors were scaled between 0 and 1. We included random intercepts for the intended category of the sketch and for each subject who participated in the guessing game; random slopes were also included for the effect of classifier evidence on each intended category.

### Crowd-sourcing semantic part labels
We designed a web-based crowdsourcing platform and recruited 50 English-speaking adult participants from Prolific to identify the basic parts of objects for each of the 16 object categories. On each trial, participants were cued with a text label of an object category and asked to list 3 to 10 object parts that came to mind (e.g., head, leg, tail, etc. for "tiger"). Participants were instructed to write only concrete parts of an object (e.g., "tail") rather than abstract attributes (e.g., "tufted"), to use common names of parts rather than technical jargon (e.g., "prehensile"), and to generate as complete a part list as they could for each object category. We applied lemmatization to the resulting part decompositions to remove redundant part labels, such as "hoof" and "hoofs", and manually edited part labels that were spelled incorrectly or with alternative spellings. We then selected the top 10% of part names that were most frequently listed. This generated a total of 82 object parts with a range of 5-13 possible parts per object category.

### Semantic part labeling

First, to ensure that these drawings were representative of the larger dataset, we chose 16 visual concepts (half animate, half inanimate) and randomly sampled drawings from children 4–8 years of age; these are the same drawings included in the visual recognition games.

We then developed a web-based annotation paradigm to obtain detailed annotations of how each pen stroke in children's drawings corresponded to the different parts of the depicted objects. 1034 English-speaking adult participants were recruited from Prolific and completed the semantic annotation task. We excluded data from 78 additional participants for experiencing technical difficulties with the web interface ($N = 11$) and for having low accuracy on our attention-check trial ($N = 67$). Data collection was stopped when every drawing had received annotations from at least three annotators.

Each annotator was presented with a set of 8 drawings randomly sampled from the drawing dataset but consistent within the same animacy and object size (i.e., small animals, large animals, vehicles, household objects). Each drawing was accompanied by the name of its object category (e.g., "airplane"), as well as a gallery of crowd-sourced part labels that corresponded to it. For each stroke in the presented drawing, annotators were prompted to tag it with the part label that described the part of the depicted object that it represented. Annotators were permitted to label a stroke with multiple part labels if they believed a stroke to represent multiple different parts of the depicted object, and were able to write their own custom label if they believed that none of the provided part labels were fitting. They could also label a stroke as unintelligible if they could not discern what it represented. Annotators also completed an "attention-check" trial, consisting of a pre-selected drawing that had been annotated by a researcher and then randomly inserted into the set of drawings. If annotators did not match the researcher's annotation criteria for this drawing, data sessions from these annotators were excluded from subsequent analysis.

### Semantic part annotation data preprocessing

First, we evaluated how often annotators agreed on what each stroke of children's drawings represented by calculating the inter-rater consistency among annotators. Across drawings, annotators agreed on the same part label for 69.9% of strokes. There was modest improvement in agreement across age, with with drawings produced by older children eliciting more consistent annotations (4-year-old drawings = 68.3% mean agreement, 8-year-old drawings = 69.8% mean agreement). We retained stroke annotations that were assigned the same part label(s) by at least two of three annotators. While annotators infrequently wrote custom labels (we did not analyze custom annotations for the present analysis) they only used 68 of the available 82 part labels. Our resultant dataset therefore contained 14,159 annotated strokes across 2088 drawings.

### Part inclusion and emphasis calculation

For part inclusion, we calculated the number of unique object parts assigned to each drawing; strokes labeled as unintelligible were not counted as distinct parts. For part emphasis, we calculated the proportion of the total length of strokes that were attributed to a particular object part in a drawing (e.g., wings), relative the total length of all strokes in the entire drawing (including strokes that were not agreed upon or that were unintelligable). If strokes were used to represent multiple object parts, we took the total length of the stroke and divided it by the number of parts that it was assigned to.

### Relating visual production and recognition

For these exploratory analyses, we used CLIP model classifications, as CLIP showed less dramatic category variation relative to VGG-19 classifications (see Supplementary Fig. 4). In addition, within the independent set of sketches used to assess children's recognition, CLIP showed a higher correlation with children's recognition behaviors (aggregating across individual sketches; VGG-19, $r = 0.28$; CLIP, $r = 0.43$).

### Reporting summary

Further information on research design is available in the Nature Portfolio Reporting Summary linked to this article.

## Data availability

All filtered drawings, raw behavioral data from the recognition experiments, and all pre-processed data that support the findings are available at https://doi.org/10.17605/OSF.IO/QYMJR, as are the pre-processed data from ref. 52.

## Code availability

The code used to analyze the data are available at the same repository as the data; https://doi.org/10.17605/OSF.IO/QYMJR.

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

## Acknowledgements

We gratefully acknowledge the San Jose Children's Discovery Museum for their collaboration and for hosting the drawing station where these data were collected. We also thank the members of the Stanford Language and Cognition Lab for their feedback on this project throughout several years and Megan Merrick for assistance with data collection. This work was funded by an NSF SPRF-FR Grant #1714726 to B.L., a NIH K99 HD108386 to B.L., and a Jacobs Foundation Fellowship to M.C.F.

## Author contributions

B.L.: Conceptualization, Methodology, Software, Formal analysis, Data Curation, Visualization, Writing—Original Draft, Writing—Review & Editing. J.E.F: Conceptualization, Software, Formal analysis, Writing—Original Draft, Writing—Review & Editing. H.H.: Software, Formal analysis, Data Curation, Resources, Visualization, Writing—Review & Editing. Z.C.: Software, Formal analysis, Data Curation, Visualization, Writing—Original Draft. M.C.F.: Conceptualization, Methodology, Resources, Supervision, Writing—Original Draft, Writing—Review & Editing.

## Competing interests

The authors declare no competing interests.
