## [Peer Review File · Nature Communications]

Reviewers' Comments:

Reviewer #1:

Remarks to the Author:

The current study looks at the development of children's object concepts, using methods of visual production and recognition. Overall, they observe that drawings become more recognizable with age (as judged by classifications made from a DNN), and as children age, they also get better at recognizing these drawings. The methods here in this study are very impressive - with the large scale of naturalistic data collected at a museum exhibit, and the DNN to quantify object drawing performance. However, I am left with some big questions of what new information these data tell us, and what this shows about the underlying representations children have for these object concepts.

Major comments

- The paper is focused around "the hypothesis that children's visual concepts change over childhood". However, it's unclear to me that this is really a meaningful hypothesis, as no one would argue the opposite. We know that infants are not born with knowledge of visual objects concepts, and we know that adults do have them, so these concepts must develop in childhood in some way. So -- these data show that these concepts develop, but we already know that they develop. What we don't know is how, or in what way / order, or from what they develop. Because of the smooth slopes in these data, it also doesn't tell us much about the developmental trajectory (e.g., so-and-so process for recognizing object concepts emerges at 4 years). I would like to see a stronger theoretical argument here in this paper. For example, is there a way to more closely compare the production vs. recognition trajectories and show that one emerges before the other? Or some way to dissect the different strategies used when making drawings by age? (See some of the later comments, too).

- I am not entirely convinced these developmental differences are purely driven by the development of visual concepts. Could any of the effects be due to language abilities that develop with age, where older children are more able to understand the prompt (for the production and recognition tasks)? Also -- to what degree could these data be about learning conventions on how draw / represent given concepts with age? Could it just be that older children are learning these norms and becoming more similar to each other in how they represent a concept (whereas younger children may have more idiosyncratic representations, or just haven't seen enough cartoon dogs to draw them)? And, older children could be better at recognizing an object from this culturally learned template. For example, I wonder if children's drawings of a rabbit actually looked like a rabbit (four legs, lowered ears), or like a cartoon representation of a rabbit (vertical ears, big eyes, on two legs - like in the 2nd column of Figure B1). Perhaps older children see more cartoons with rabbits and learn the strategies in drawing them. In contrast, younger children could still recognize a rabbit or a cat when they see it in the world, but just not have yet learned the cultural norms for how to draw them. I see in Figure 2 that many of the worst classified categories by the DNN seem to be those where a culturally-learned template is different from the actual appearance of the object (e.g., many of the animals). Overall, these results may indicate just that children are learning cultural norms over time, or accumulating experiences of how these objects are drawn in the world. This seems fundamentally different from the development of visual concepts divorced from a drawing task.

- The discussion suggests a possible account that "children are explicitly learning the diagnostic features of categories as they enrich their semantic knowledge". I think testing these specific accounts would be key to understanding what is actually happening during development, and with the large amount of data it should be possible. What features / object parts are driving the successful categorization of these categories with age? For example, you could try and see what features the DNN is using to make its successful classifications. Or, you could see what features exist or emerge with age and see whether people can still identify the drawings if those features are removed.

Minor Comments

- "These models have no knowledge of drawing conventions" -- However it seems like the DNN was re-trained on drawing data in order to make the classification. It's thus not clear to me this DNN is not learning some mapping of certain contours = a specific object (rather than: this drawing really resembles a true cat).
- Did you test for multicollinearity of the factors in your models - to ensure that none of these factors were highly correlated?
- What happens with the recognition phase data when you include the age of the drawer in the model? I am curious - is there any sort of interaction (perhaps children can better recognize drawings from their same age group)? Also, since you have recognition data here, I'd like to see if these human judgments replicate the results of the DNN. Are drawings made by older children better recognized? And if not -- does this indicate something strange about what the DNN is doing (e.g., it's instead picking up on consistent templates developed by older children to draw a given concept)?
- Could you provide more information on the selection of the 48 object categories?
- It seems that this work should cite some of the drawing studies from Wilma Bainbridge that also look at drawings and people's ability to recognize them (e.g., Bainbridge et al., 2019. Nature Communications).
- For Figure 3A, it would be helpful to mention or indicate the order the categories are listed for the matrix - I assume the authors intend for the reader to see the lighter spots separately encompassing animate objects and inanimate objects. Currently, it's not clear how the reader is intended to interpret that figure.
- Is it surprising that the ability to convey the correct animacy and size category doesn't seem to change with age?
- "Children became steadily better at identifying the category that a drawing referred to" -- this sounds like it means a given child got better over time at recognizing the images - was this the intention of the sentence?
- Table 2 should include p-values as already present in Table 1.
- Can you provide more information about the manual filtering of the drawings? This is just to ensure that there were no biases in how information was filtered out that may be related with age.
- I see quite a spread in performance -- some young children outperformed some of the older children. Have you looked at any individual difference metrics - like are there certain groups of high-performing vs. low-performing participants that show interesting differences in their drawings?

Reviewer #2:

Remarks to the Author:

This manuscript reports work on the developmental relation between children's graphic productions and children's visual concept knowledge. Strengths of the manuscript include the very large dataset that was collected, involving semi-automated procedures at a children's museum, and the creative use of machine learning procedures to recognize and classify children's drawings into basic level categories. Despite these strengths, the more theoretical and substantive contributions of the work are less certain.

Some of the work is framed by hypotheses that come off either as overly simplistic or that have already appeared in the literature. With regard to the former issue, the hypothesis that pictorial competence (e.g., interpreting pictures, drawings as depictions of objects) either appears early in development or develops gradually during childhood seems reductionistic. In fact, the authors

settle on the idea that competence in graphic productions (i.e., the use of diagnostic features in drawings) and competence in visual conceptual knowledge drive each other during development. This is an interesting idea, but the hypothesis has already been advanced and supported in the literature (some of which is reviewed by the authors), with more direct experimental methods than those reported here.

This issue brings me to my second general concern with the investigation. Although the machine learning approach is valuable and advances the developmental literature on children's drawings, the experimental design here is largely correlational. Separate samples are studied to investigate a) children's drawings, and b) children's ability to classify or identify these drawings. The unsurprising conclusion that emerges is that children get better at both skills with age. But to understand how children's external and internal representations of objects are related developmentally, a within-subject design would be more appropriate, or relatedly, some type of training study, where children who are given training in drawing are then tested on their visual classification abilities and/or vice versa. Indeed, this type of approach has already been used in the literature (and again, some of this work is referenced by the authors), allowing conclusions that are more causal.

Finally, I believe the machine learning approach adopted by the authors to investigate these developmental questions has great potential, and perhaps should be written up for a more discipline-specific journal in developmental science, but some issues involving this approach require additional clarification. When the authors queried adults whether they helped their children draw, the results indicated that "Out of 11797 subsequent sessions at the station, 3094 filled out the survey, and 719 reported interference, 6.09% of participants; ...". If I am interpreting this correctly, however, the rate of interference for those who returned the survey is closer to 23%, which leads to concerns about whether a substantial number of children completed the drawings on their own.

Additionally, the authors do not discuss the issue of non-independence within the dataset: the possibility that the same children may have contributed data at different points in time during the period when the museum kiosk was available.

Lastly, in the drawing classification task, children had to match a drawing to one of a subset of photographs. Performance here was compared to a control photograph matching task, in which children had to match a photograph to another photograph "that goes with the picture". However, any difficulties that the children, especially the younger ones, may have had when matching a drawing to a photograph, may have been due to the different media (drawing vs photograph) involved. In other words, the experimental design here is not fully balanced, and there is a confound of sorts. Needed as well is another control task in which children must match a drawing to another drawing.

Reviewer #3:

Remarks to the Author:

Key results: The manuscript „Parallel developmental changes in children's drawing and recognition of visual concepts" reports findings on the combination of an elegantly designed behavioral study with children aged 2-10 years and the use of a deep convolutional neural network (DCNN) model of object recognition. Based on >37.000 drawings of children, the study reveals a protracted development in the use to include diagnostic visual features in drawings as well as in using these features to recognize the drawings of others.

Significance and recommendation: This is a most valuable contribution to the research on human visual concept development based on a large amount of data and using sophisticated analyses techniques. I am confident that this will inspire more "field" research and hence further deepen our understanding of human development. Overall, I think this is an excellent study that definitely merits publication in Nature Communications and recommend accept pending minor revisions (see below). Congratulations!

My expertise: I am no expert in DCNN and hence will not comment on this methodological aspect but rather focus on the other aspects with regard to the behavioral study.

Validity / Quality of the data / Data and methodology: The findings are based on a most cleverly designed behavioral study with a lab space inside a local science museum to record an astonishing number of drawings and recognition of drawings by children. Authors took great care to assure data quality and I firmly believe that such measurements "in the wild" – if they are based on a large N as is the case here – tell us more than stringent laboratory-based testing with a small N. Data is analyzed and interpreted carefully and presented in sufficient detail. I found the control conditions – e.g. the tracing assessment, analysis with regard to frequency, animacy ect. – especially thoughtful.

Analytical approach: I find the conclusions strongly supported by the empirical evidence, especially since authors took great care in accounting for potential other interpretations (e.g. with regard to unrecognizable drawings).

Suggested improvements: One aspect that I did not quite follow is an argument of the discussion: Page 16 "Using this approach, we find evidence for continuous and variable changes in children's visual concepts across development – rather than a point at which children become "adult-like". We believe that this work paints a more accurate picture of developmental change and opens up new avenues for investigating the various factors that shape visual concepts throughout development." I understand that a broad age range and many data points enable us to see the developmental trajectory with a high precision and that this might be the way to go forward. However, at the same time it would still be interesting to investigate adults with the same task and to see, how they perform – completely independent from arguing if and when a behavior becomes "adult-like" or whether this even makes sense. I have the feeling that the current data set has no evidence for criticizing other research for investigating "adult-like" behavior when in fact it did not even include adults.

Clarity and context: I find the manuscript clearly written and all information provided accessible.

References: The manuscript references previous literature appropriately including hard-to-find literature on the development of drawing abilities.

Reviewer #4:

Remarks to the Author:

Deep learning and data-science are rapidly reshaping science. This paper showcases how DNNs in combination with a large database of images can be used in developmental studies.

Children improve in recognising objects with age. The authors try to differentiate between a) the hypothesis that children's visual concepts change gradually over time as they learn which features are most diagnostic, and b) these concepts do not change, but that improvement is a result of specific experiences or better control.

In this paper the authors equate 'amount of diagnostic features' in an image with the performance of a DNN on this image. In this way they are able to sidestep having to develop intuitions on visual complexity. By combining this with a large data-set of images, produced and labelled by children in the range of 2 to 10 years old .

The paper is inspiring and ambitious in its goals but I would say that the main conclusion, while plausible, are not entirely convincing.

Major

- The authors have, for a substantial part of their paper, a strong quantitative approach. This is less so for distinguishing between hypothesis a and b. Would it be possible to specify the outcomes of each of these hypothesis in a quantitative way (for instance via an RDM or other type of prediction). This would make it possible to see what is the better model and to what degree. With

the current setup the main conclusions are somewhat narrative.

o Currently the FC1 model is trained with data from all age ranges. What happens if you would train with age range 2-4 and age range 8-10? If I understand the reasoning this is not expected to be symmetrical? I think the authors would predict 8-10 would classify, on average 2-4 better, than vice versa. Is this correct? Hypothesis b would not predict this?

o As an example of a narrative conclusion, on page 12 the authors state that the children's ability to control and plan their motor movements, but does not entirely account for. To what degree is does and does not?

- Much depends on the assumptions that FC1 of VGG can be used to evaluate the amount of diagnostic features in an image. I think it is therefore paramount that the authors show that a similar pattern is not present for Conv1 and Conv2, as a control. It would also be interesting to see to what degree this develops over the layers, and how/if this relates to development.

Minor

- p8, it seems like a weak control to ask parents how often children draw each category, although better than alternatives. Would it be also possible to relate this to a popular source of frequencies of words? This might tap into the exposure to a certain category.

- A change that is not covered by the study, but is referenced to, is the context of an object in relation to other objects. This is not covered by the drawings of single objects. Any gains on this basis cannot be evaluated.

Disclaimer: I have no specific background in developmental studies, mainly vision and DNNs

Reviewer Comments

Reviewer #1

The current study looks at the development of children's object concepts, using methods of visual production and recognition. Overall, they observe that drawings become more recognizable with age (as judged by classifications made from a DNN), and as children age, they also get better at recognizing these drawings. The methods here in this study are very impressive - with the large scale of naturalistic data collected at a museum exhibit, and the DNN to quantify object drawing performance. However, I am left with some big questions of what new information these data tell us, and what this shows about the underlying representations children have for these object concepts.

Major comments

- The paper is focused around "the hypothesis that children's visual concepts change over childhood". However, it's unclear to me that this is really a meaningful hypothesis, as no one would argue the opposite. We know that infants are not born with knowledge of visual objects concepts, and we know that adults do have them, so these concepts must develop in childhood in some way. So -- these data show that these concepts develop, but we already know that they develop.

Thanks for these comments, which were very helpful to us. The revised manuscript now clarifies our theoretical contribution and hypotheses via several new analyses and a revised introduction. Instead of asking whether or not visual concepts change across childhood, we now examine *how* they change.

We have also clarified a possible alternative hypothesis. Indeed, one real possibility was that the bulk of learning about "what things look like" might have been more or less completed by the end of the preschool years—by around age 4, when children produce language relatively fluently and show similar visual recognition abilities to adults in some ways (including in our own prior work). Anecdotally, many researchers had the intuition that we would not find changes in how well children could recognize drawings of object categories – on this account, changes in children's drawings only reflect changes in how children can plan and control their motor movements. Indeed, it seems that many toddlers can readily recognize the *rabbits* both in picture books and in the real-world – and so many made the prediction that we wouldn't find developmental changes in this recognition task. Contra this intuition, however, we found a long, smooth developmental trajectory throughout the entire age range we tested.

A related possibility was perhaps that we would see a plateau in developmental change around ages 6, when children begin to write and can much more readily control how they move a pencil or a pen. However, we still see developmental gains even in these later years. So, we think that our findings run contra both of these intuitive possibilities and are not predicted by accounts of visual development that posit that the bulk of visual category learning has been accomplished by the preschool years. In the revised manuscript, we now detail this alternative account and specific predictions in the introduction.

Instead, we suggest that children gradually learn more about the diagnostic features of visual concepts throughout early and middle childhood. As children include and emphasize diagnostic

features of object categories in their drawings, they rely on those very same features when recognizing drawings of object categories. In the revised manuscript, we now more clearly situate our hypotheses and findings with respect to these theoretical accounts and intuitions.

What we don't know is how, or in what way / order, or from what they develop. Because of the smooth slopes in these data, it also doesn't tell us much about the developmental trajectory (e.g., so-and-so process for recognizing object concepts emerges at 4 years). I would like to see a stronger theoretical argument here in this paper. For example, is there a way to more closely compare the production vs. recognition trajectories and show that one emerges before the other? Or some way to dissect the different strategies used when making drawings by age? (See some of the later comments, too).

Again, thank you for the helpful prompt!

To make progress on the “how” question, we now include new analyses that (1) examine changes in how children both include object parts (e.g., “ears”) in their drawings across development, and (2) examine how the presence of different object parts affects children’s recognition of each other’s drawings.

To do so, we collected a set of part-based annotations of the drawings, allowing us to examine which object parts are present in children’s drawings over development as well as how the inclusion and emphasis of different parts (e.g., *wheels, wings*) changes over development (see Figures 4,5). These findings ground out the deep neural network classifications in interpretable changes in drawings constituent parts, allowing insight into how the drawings are changing and into what makes a given visual concept more or less recognizable. So, our revised manuscript also now starts to make progress on understanding what these diagnostic features could actually be.

Finally, while the present data document changes in visual production and recognition of drawings across childhood, the reviewer is right to note that it cannot directly examine the relationship between these two abilities beyond examining variation at the item level. In the revised manuscript, we now include an exploratory analysis relating production vs. recognition at the category level, leveraging the larger dataset to relate recognition and production for independent sets of drawings. We find modest correlation between which categories tend to be better produced and recognized at each age, further pointing towards the idea that common representations may underlie children’s performance in both tasks (see Figure 7).

Future work that uses either longitudinal or within-subjects design may be able to assess the degree to which visual recognition abilities might somewhat precede visual production – or vice versa. In the revised manuscript, we now more clearly lay out this limitation in the General Discussion and detail the different possibilities for how these two abilities may interact across development. We hope that the open dataset and methods advanced by this paper will spur work on this topic.

- I am not entirely convinced these developmental differences are purely driven by the development of visual concepts. Could any of the effects be due to language abilities that develop with age, where older children are more able to understand the prompt (for the production and recognition tasks)?

The developmental differences we observed are almost certainly driven by many more factors than just the development of visual concepts – task engagement, drawing effort, visuomotor control (among other cognitive abilities) – also change across childhood.

Across both tasks, we attempted to account for these additional factors in several ways. First, to ensure that the youngest children understood these tasks, we conducted iterative, in-person piloting sessions before installing both the visual production and visual recognition experiments at the museum kiosk. We were convinced from these in-person sessions that they understood the task at hand. In addition, the museum kiosk was designed such that children could not start the session on their own before an adult caregiver responded to the consent prompts and entered their child’s age, ensuring that they were present for the beginning of the session.

Second, in the visual production task, we measured task engagement via effort covariates (time spent, ink used, and strokes made) as well tracing performance, and accounted for these factors in our mixed-effect models. In the recognition task, we ensured that children were on task by only including children who performed >75% on the catch trials, excluded 2-year-olds from the dataset (who may not understand this kind of forced choice task), and also conducted a set of analysis where we only included children who performed at ceiling on the catch trials, replicating the main results.

Third, in the revised manuscript, we now report a replication of the main visual production results during a controlled experimental session where an experimenter was present (N=121 children, 16 categories, ages 4-9 years) with a more limited set of categories (see Appendix, Figure A4).

In sum, while we acknowledge that there are many other developmental changes occurring during the period we tested – a two-year-old is not anywhere close to a ten-year-old on many dimensions in their abilities! – we do not believe that the data support a large explanatory role for effort, engagement, or task comprehension.

Also -- to what degree could these data be about learning conventions on how to draw / represent given concepts with age? Could it just be that older children are learning these norms and becoming more similar to each other in how they represent a concept (whereas younger children may have more idiosyncratic representations, or just haven't seen enough cartoon dogs to draw them)? And, older children could be better at recognizing an object from this culturally learned template. For example, I wonder if children's drawings of a rabbit actually looked like a rabbit (four legs, lowered ears), or like a cartoon representation of a rabbit (vertical ears, big eyes, on two legs - like in the 2nd column of Figure B1). Perhaps older children see more cartoons with rabbits and learn the strategies in drawing them. In contrast, younger children could still recognize a rabbit or a cat when they see it in the world, but just not have yet learned the cultural norms for how to draw them.

Thanks for this great set of questions. Learning conventions is likely a part of what we are observing in our data, but surprisingly, more frequently drawn categories are not actually more recognizable (as our results section with the same name shows).

More generally, different cultures may tend to depict certain categories in particular ways, e.g., exaggerating certain facial features (e.g., eyes) in animals, or using a very sparse line to depict a “bird”. We think of these culturally specific differences as part and parcel of our visual

concepts: we learn to identify and imagine *birds* in drawings, illustrations, videos, as toys, or in real-life, all of which demands extraordinarily flexible and rich representations. As a result, we think that visual concepts for “rabbits” may really vary among adults in different cultural contexts who depict them differently – and, for that matter, quite differently in cultures where children and adults consume and produce relatively few depictions overall (e.g., some rural, aboriginal contexts).

So, we agree with the reviewer more broadly that part of what may be changing across development is the degree to which these different visual cultural conventions are incorporated into children’s visual concepts, and perhaps children learn these conventions through producing drawings themselves. We think that this is likely part of the story, but not all of it: we still observed substantial changes over development in both production and recognition for rarely drawn categories (e.g., *clocks, scissors, bottle*) that tend to be less frequently drawn by children – and thus presumably to have less iconic and presumably less culturally variable representations (though this is an active area of investigation, see Lewis et al., 2021).

In the revised manuscript, we discuss the degree to which we expect to find cultural variation across the trends we observed in the General Discussion.

I see in Figure 2 that many of the worst classified categories by the DNN seem to be those where a culturally-learned template is different from the actual appearance of the object (e.g., many of the animals). Overall, these results may indicate just that children are learning cultural norms over time, or accumulating experiences of how these objects are drawn in the world. This seems fundamentally different from the development of visual concepts divorced from a drawing task.

Thanks for this thought-provoking observation. A more general form of the question might be, “What drives how well children can produce and recognize a drawing of a given category (e.g., a dog), and how related are these improvements to children’s changing internal visual concepts?” In the revised manuscript, we have taken several steps to try to understand and model category-level variation.

First, we applied an additional deep neural network recognition model to our drawing dataset – CLIP, or Contrastive Language–Image Pre-training, see <https://openai.com/blog/clip/>) which is trained on the correspondences between images and captions and as of 2022 is now state-of-the-art for zero-shot recognition of different visual concepts. In addition, CLIP outperforms other convolutional-neural networks in recognition tasks that vary across visual formats (e.g., as sketches), and its embeddings have been used in sketch-generation models (Vinker et al., 2022), making it a natural choice for a second model to use as a robustness check.

We then used these two model outputs to examine the consistency and variability in category-level effects. Overall, we found that the two models exhibited some consistency in item effects (see Figure) as well as some variabilities: while CLIP (vs VGG) was better at recognizing drawings of small animals (including dogs), “dogs” were still a poorly recognised category overall; other categories (e.g., *hand, spider, couch, frog*) were equally well-recognized between the two models. However, VGG still performed considerably better at recognizing some of the infrequently drawn categories (e.g., *TV, piano*).

Figure. Comparing category-level classification variation between VGG-19 (y-axis) vs. the CLIP model (x-axis) on the large observational dataset.

We then modeled the item variation in both CLIP, VGG, and human recognition (from the new, experimental dataset with 12 categories) – as a function of (1) the frequency with which parents estimate their children draw these categories, and (2) the frequency with which this word appears in the English books (from Google N-grams, following a suggestion from reviewer #3, and (3) following the earlier concern, when this word tends to be acquired in development, as estimated by the MacArthur Bates CDI norms via the Wordbank database (wordbank.stanford.edu).

None of these factors explained significant variability in any of the item effects that we observed (see Appendix, Table A5).

So, we suspect that these category-level effects are likely due to the interaction of several different factors, including (but not limited to) (1) the perceptual similarities between the categories in the dataset (e.g., *dog* vs. *rabbit* are more confusable with each other than they are with *hands*), (2) the visuomotor demands of accurately depicting a given category (*airplanes* may be more difficult to depict than *apples*), and (3) for model classifications, any idiosyncrasies in the training regimes and the resulting visual feature spaces.

Future work may also test (4) the degree to which drawings of categories vary across cultures (Lewis et al., 2021) or (5) the degree to which recognizable drawings resemble photographs of their real-world counterparts (Vinker et al., 2022) are related to how well children can produce and recognize a given category across development. While deriving accurate measures of cultural variability and iconicity is beyond the scope of the present paper, we expect that both of these factors may explain some variability in how children produce and recognize visual concepts across development.

The discussion suggests a possible account that "children are explicitly learning the diagnostic features of categories as they enrich their semantic knowledge". I think testing these specific accounts would be key to understanding what is actually happening during development, and with the large amount of data it should be possible. What features / object parts are driving the successful categorization of these categories with age? For example, you could try and see what features the DNN is using to make its successful classifications. Or, you could see what features exist or emerge with age and see whether people can still identify the drawings if those features are removed.

We agree that an understanding how features / object parts are related to changes in recognition is critical to this endeavor.

In the prior version of the manuscript, we were not able to address this question. DNN feature activations tend to be hard to interpret: identifying which of thousands of FC6 features are relevant for *birds vs. dogs* is unlikely to produce meaningful results. And object drawings contain explicit object-part structure in a way that photographs of visual concepts do not—and prioritize the relevant object parts for recognition.

In the revised manuscript, we now address this question more directly. First, we now directly examine how the inclusion and emphasis of different object parts (e.g., *wheels, eyes*) vary across development and contribute to successful DNN classifications, allowing us to quantify, for each category, the different object parts that change across development and lead to a more or less recognizable exemplar (see Appendix, Figure A9). Overall, we found that some categories had a clear object part (e.g., *wool* for *sheep*) whose inclusion and emphasis changed over development and tended to lead to better recognition. For other categories, however, it was more difficult to identify a singular object part that was ‘diagnostic’ of a different category; rather, it appears that the inclusion and emphasis of multiple different object parts is changing (e.g., *elongated ears* relative to a head for *rabbits*). Thus, these novel results provide qualitative insight into the relevant changes in children’s drawings across development.

Second, our new part annotations and related analyses (Figures 4,5) provide some exploration of the importance of particular object parts for children of different ages. While the picture is clear for some categories (e.g., the presence of a “wheel” for *car* increases with age), it is less clear for many other animate categories (e.g., even younger children’s unrecognizable drawings of *rabbits* often contain *ears*).

Minor Comments

- "These models have no knowledge of drawing conventions" -- However it seems like the DNN was re-trained on drawing data in order to make the classification. It's thus not clear to me this DNN is not learning some mapping of certain contours = a specific object (rather than: this drawing really resembles a true cat).

The reviewer is correct that a logistic regression classifier is trained on top of the VGG model feature activations — and that it is not classifying “cat drawings” as “cats”; rather, it is learning to associate the set of high-level perceptual features that tend to be diagnostic of the different

categories in this dataset. In the revised manuscript, we have clarified how we are using the DNN activations in the main text and revised this sentence in the introduction.

Further, CLIP (the new state-of-the-art model that we include) uses a much larger corpus of training and is very likely more robust to image style variation (e.g., picture vs. drawing).

- Did you test for multicollinearity of the factors in your models - to ensure that none of these factors were highly correlated?

Some of our predictors could have reasonably been correlated – for example, the number of strokes used and “ink” used have both been used as indexes of drawing effort in prior work. However, the correlations between those predictors are relatively low (all r 's < .3). In more systematic testing, we found that the VIF (variance inflation factor) for all of our predictors was less than 2, indicating relatively low to moderate severity. This multicollinearity analysis is now included in the Appendix (see Table A2).

- What happens with the recognition phase data when you include the age of the drawer in the model? I am curious - is there any sort of interaction (perhaps children can better recognize drawings from their same age group)?

Initially, we too had this intuition, thinking that perhaps younger children are better at recognizing other drawings made by younger children (it would be interesting if they had their own conventions not shared with adults or older children!). However, we didn't find any evidence for this idea. Instead, it seems that drawings by younger children are just less recognizable overall (see next response).

Also, since you have recognition data here, I'd like to see if these human judgments replicate the results of the DNN. Are drawings made by older children better recognized? And if not -- does this indicate something strange about what the DNN is doing (e.g., it's instead picking up on consistent templates developed by older children to draw a given concept)?

In the recognition data, we replicate the main trends seen in the larger dataset: drawings by older children are better recognized (by children of all ages). In the revised manuscript, we've added this additional plot to the Appendix, following up on the above comment, which plots the interaction between producer age and recognizer age (see Appendix, Figure B2).

- Could you provide more information on the selection of the 48 object categories?

We have added additional details to the methods section. In brief, we chose sets of 8 categories that included both animate/inanimate items, were a variety of frequently/infrequently drawn categories, and preferentially chose items that were also in the QuickDraw database to facilitate uptake of these data by other groups.

- It seems that this work should cite some of the drawing studies from Wilma Bainbridge that also look at drawings and people's ability to recognize them (e.g., Bainbridge et al., 2019. Nature Communications).

We agree that this work should be cited; thanks for bringing this to our attention.

- For Figure 3A, it would be helpful to mention or indicate the order the categories are listed for the matrix - I assume the authors intend for the reader to see the lighter spots separately encompassing animate objects and inanimate objects. Currently, it's not clear how the reader is intended to interpret that figure.

We thank you for the attention to detail; we have revised the figure and added additional information about the animacy/real-world sizes of the categories.

- Is it surprising that the ability to convey the correct animacy and size category doesn't seem to change with age?

Prior work has found that 3-4 year-olds process the animacy/size of objects they cannot recognize in both visual search and stroop tasks (Long et al., 2019 *JEP:HPP* , Long et al. 2019 *Visual Cognition*). So, we expected that children might be able to use simple curvature features to convey this animacy/size equally throughout development, even if they were unable to produce recognizable drawings that are distinguished at the basic-level. To clarify, this analysis was conducted only on the misclassified drawings (vs. the entire set of recognizable and misclassified drawings). In the revised manuscript, we have clarified this analysis.

- "Children became steadily better at identifying the category that a drawing referred to" -- this sounds like it means a given child got better over time at recognizing the images - was this the intention of the sentence?

This was not our intention, as this was not a longitudinal study. Thanks for alerting us to this ambiguity; we have revised this sentence.

- Table 2 should include p-values as already present in Table 1.

Thanks for catching this; we have added them.

- Can you provide more information about the manual filtering of the drawings? This is just to ensure that there were no biases in how information was filtered out that may be related with age.

The manual filtering of the drawings was conducted by one of the authors viewing the drawings in the dataset; while filtering was not done within age groups, the author was not blinded to this information. Thus, one concern is that the author could have been biased to include more unrecognizable drawings in the younger vs. older age ranges, which could artificially create an age effect. If this was the case, this bias would predict lower exclusion rates in the youngest age groups for drawings that had been manually filtered vs. filtered by online participants.

However, we did not find this was the case: filtering rates were overall much higher by participants in online experiments and do not appear disproportionately lower in this younger age group for manual exclusions. We expected this pattern of results, as younger children produce scribbles more often than older children, and to be conservative we excluded any drawing that was marked invalid by a participant.

Figure. Y-axis shows the percent of drawings that were excluded within each age group during the behavioral, online experiment (administered via Prolific) vs. excluded by an author manually examining the drawings. The size of the dots represents the number of drawings examined in each age group.

- I see quite a spread in performance -- some young children outperformed some of the older children. Have you looked at any individual difference metrics - like are there certain groups of high-performing vs. low-performing participants that show interesting differences in their drawings?

In Figure 2A, the individual dots represent the 48 categories (rather than data from individual children); we have clarified this convention in the Figure. Nonetheless, the reviewer is indeed correct that there was quite the spread in performance across children (and even among children who all received the same items). We found that a large portion of variance is explained by children's individual performance on the tracing trials. In the revised manuscript, Table 1 shows this effect, relating individual tracing scores to drawing recognition and quantifying the relative strength of tracing vs. children's age as predictors; predictors have now been z-scored so that coefficient magnitudes are comparable.

Reviewer #2

This manuscript reports work on the developmental relation between children's graphic productions and children's visual concept knowledge. Strengths of the manuscript include the very large dataset that was collected, involving semi-automated procedures at a children's museum, and the creative use of machine learning procedures to recognize and classify children's drawings into basic level categories. Despite these strengths, the more theoretical and substantive contributions of the work are less certain.

Some of the work is framed by hypotheses that come off either as overly simplistic or that have already appeared in the literature. With regard to the former issue, the hypothesis that pictorial competence (e.g., interpreting pictures, drawings as depictions of objects) either appears early in development or develops gradually during childhood seems reductionistic. In fact, the authors settle on the idea that competence in graphic productions (i.e., the use of diagnostic features in drawings) and competence in visual conceptual knowledge drive each other during development. This is an interesting idea, but the hypothesis has already been advanced and supported in the literature (some of which is reviewed by the authors), with more direct experimental methods than those reported here.

Thank you for raising this issue.

In the revised manuscript, we more fully lay out the prior developmental and adult work that has addressed the question of gradual development of visual concepts. Our specific hypothesis is that production and comprehension develop in parallel, and – to our knowledge – the tests of this hypothesis are quite tentative at present, due to: (1) the lack of parallel observation of production and recognition, (2) the small sample of visual categories that has been investigated, and (3) the small developmental range being studied. If there are further references that the reviewer can provide, we would be happy to incorporate them in the manuscript.

In this context, the novel contributions of the current investigation are (1) to provide data from a much larger sample of categories, participants, and age-ranges using novel computational analyses and (2) provide comparable recognition judgments for the same categories, thus measuring visual production and recognition in parallel.

This issue brings me to my second general concern with the investigation. Although the machine learning approach is valuable and advances the developmental literature on children's drawings, the experimental design here is largely correlational. Separate samples are studied to investigate a) children's drawings, and b) children's ability to classify or identify these drawings. The unsurprising conclusion that emerges is that children get better at both skills with age. But to understand how children's external and internal representations of objects are related developmentally, a within-subject design would be more appropriate, or relatedly, some type of training study, where children who are given training in drawing are then tested on their visual classification abilities and/or vice versa. Indeed, this type of approach has already been used in the literature (and again, some of this work is referenced by the authors), allowing conclusions that are more causal.

Thanks for bringing up this issue. It is certainly correct that our approach is correlational (as a longitudinal or within-subjects design would be as well). However, a large-scale study to understand variation in drawing abilities is an important prerequisite to further experimental work such as the proposed training study. Further, we use item-variation (rather than participant-level variation) as a lever to overcome this issue; by trying to understand which items are produced and recognized more and less well we can gain insight into the factors that do – and in our case, do not – explain this variation (see e.g. Braginsky et al., 2019 for discussion of this strategy).

In the revised manuscript, we now highlight the limitations of the correlational nature of this dataset and the opportunities for future work in the General Discussion, saying clearly that *“within-child measurements will be necessary to confirm the hypothesis that changes in children's visual concepts underlie the observed changes in both tasks.”*

As above, we would appreciate any references to training studies in the literature that link production and recognition; we are only aware of work on letter recognition and not on visual concepts more broadly.

Finally, I believe the machine learning approach adopted by the authors to investigate these developmental questions has great potential, and perhaps should be written up for a more discipline-specific journal in developmental science, but some issues involving this approach require additional clarification. When the authors queried adults whether they helped their children draw, the results indicated that “Out of 11797 subsequent sessions at the station, 3094 filled out the survey, and 719 reported interference, 6.09% of participants; ...”. If I am interpreting this correctly, however, the rate of interference for those who returned the survey is closer to 23%, which leads to concerns about whether a substantial number of children completed the drawings on their own.

During piloting, we observed that many children completed the task on their own after their caregiver helped them advance past the consent screen. As the survey was optional and appeared at only the end of the task (if they pressed “stop” or completed all 8 trials), we anticipated that there were likely to be many children whose parents didn’t complete the survey because they were not present while their child was participating (while observing, this was often the case that their caregiver was attending to another child or chatting while their child completed the drawing study). Thus, this imperfect measure likely only captured responses from parents or older children who were actually present and watching their child draw.

However, we too were worried that perhaps these age-specific trends could be changed in some way by either interference from parents/other children or the DNN measures that we used. We thus replicated these results in an experimental, controlled session where we could eliminate all sources of interference, and we then had adults (via Prolific) attempt to recognize these drawings. With this controlled dataset, we saw the same pattern of results in this smaller dataset with 16 categories and 120 participants (4-9 years of age): the recognizability of the drawings children produced increased steadily with age and we did not see any effect of drawing frequency. We now report these results in the Appendix as validation of our main finding and of the model classifications.

Additionally, the authors do not discuss the issue of non-independence within the dataset: the possibility that the same children may have contributed data at different points in time during the period when the museum kiosk was available.

It is true that the same children could have contributed data multiple times; we do not have any way of quantifying this non-independence. In practice, since the categories being used in the drawing and recognition tasks were rotated, a child completing the task multiple times might simply have contributed drawings or judgments for more of the possible categories. During the two days that we observed the museum kiosk, we did not notice children completing the task more than once in a row.

Given the large size of our dataset, a small number of repeat participants would not have an appreciable effect on our statistical results. In the revised manuscript, we now clarify that some children could have participated multiple times.

Lastly, in the drawing classification task, children had to match a drawing to one of a subset of photographs. Performance here was compared to a control photograph matching task, in which children had to match a photograph to another photograph “that goes with the picture”. However, any difficulties that the children, especially the younger ones, may have had when matching a drawing to a photograph, may have been due to the different media (drawing vs photograph) involved. In other words, the experimental design here is not fully

balanced, and there is a confound of sorts. Needed as well is another control task in which children must match a drawing to another drawing.

Thanks for bringing up this issue, which we thought through ourselves during the design phases. In designing our recognition task, we needed some way of measuring visual recognition in children spanning a wide age range, some of whom could not read. We chose pictures as our “labels” for the matching task, following the extensive literature suggesting that even infants and very young children have no trouble matching drawings and pictures (e.g., Deloache et al., 1971; Hochberg & Brookes, 1962).

If younger children had difficulty matching drawings to a photograph button, this effect could artificially reduce the performance of younger children on this task and thus create an age effect. To rule out this explanation, we analyzed the subset of children who showed both (1) high overall drawing recognition performance and (2) ceiling performance on the photograph matching catch trials. This subset still showed differences across age groups – and, critically, the key interaction with classifier evidence. These analyses are detailed in the Appendix (see Table B1).

As a final check, we assessed qualitatively whether younger children were disproportionately worse at recognizing object categories that had more iconic representations. For example, the photo icon for a “boat” was a motorboat – whereas many children drew boats with sails (see part annotations section, Figure 5). Contra the idea that picture matching would be challenging in this case, *boats* were the best recognized category by children – both overall and even at the youngest age (see item effects and the corresponding picture “labels” in Appendix Figure B3).

In the revised manuscript, we have highlighted this concern and these additional analyses in both the appendix and the main text.

Reviewer #3

Key results: The manuscript „Parallel developmental changes in children’s drawing and recognition of visual concepts” reports findings on the combination of an elegantly designed behavioral study with children aged 2-10 years and the use of a deep convolutional neural network (DCNN) model of object recognition. Based on >37.000 drawings of children, the study reveals a protracted development in the use to include diagnostic visual features in drawings as well as in using these features to recognize the drawings of others.

Significance and recommendation: This is a most valuable contribution to the research on human visual concept development based on a large amount of data and using sophisticated analyses techniques. I am confident that this will inspire more “field” research and hence further deepen our understanding of human development. Overall, I think this is an excellent study that definitely merits publication in Nature Communications and recommend accept pending minor revisions (see below). Congratulations!

My expertise: I am no expert in DCNN and hence will not comment on this methodological aspect but rather focus on the other aspects with regard to the behavioral study.

Validity / Quality of the data / Data and methodology: The findings are based on a most cleverly designed behavioral study with a lab space inside a local science museum to record an astonishing number of drawings and recognition of drawings by children. Authors took great care to assure data quality and I firmly believe that

such measurements “in the wild” – if they are based on a large N as is the case here – tell us more than stringent laboratory-based testing with a small N. Data is analyzed and interpreted carefully and presented in sufficient detail. I found the control conditions – e.g. the tracing assessment, analysis with regard to frequency, animacy ect. – especially thoughtful.

We thank the reviewer for their kind comments.

Analytical approach: I find the conclusions strongly supported by the empirical evidence, especially since authors took great care in accounting for potential other interpretations (e.g. with regard to unrecognizable drawings).

Suggested improvements: One aspect that I did not quite follow is an argument of the discussion: Page 16 “Using this approach, we find evidence for continuous and variable changes in children’s visual concepts across development – rather than a point at which children become “adult-like”. We believe that this work paints a more accurate picture of developmental change and opens up new avenues for investigating the various factors that shape visual concepts throughout development.” I understand that a broad age range and many data points enable us to see the developmental trajectory with a high precision and that this might be the way to go forward. However, at the same time it would still be interesting to investigate adults with the same task and to see, how they perform – completely independent from arguing if and when a behavior becomes “adult-like” or whether this even makes sense. I have the feeling that the current data set has no evidence for criticizing other research for investigating “adult-like” behavior when in fact it did not even include adults.

We agree that this idea should be clarified, and we have now removed this sentence from the Discussion and reworded this section. In particular, we have added material in the Introduction highlighting that adults may vary in the specificity of their visual concepts, visual expertise, and resulting drawings.

Clarity and context: I find the manuscript clearly written and all information provided accessible.

References: The manuscript references previous literature appropriately including hard-to-find literature on the development of drawing abilities.

Reviewer #4

Deep learning and data-science are rapidly reshaping science. This paper showcases how DNNs in combination with a large database of images can be used in developmental studies.

Children improve in recognising objects with age. The authors try to differentiate between a) the hypothesis that children’s visual concepts change gradually over time as they learn which features are most diagnostic, and b) these concepts do not change, but that improvement is a result of specific experiences or better control.

In this paper the authors equate ‘amount of diagnostic features’ in an image with the performance of a DNN on this image. In this way they are able to sidestep having to develop intuitions on visual complexity. By combining this with a large data-set of images, produced and labelled by children in the range of 2 to 10 years old .

The paper is inspiring and ambitious in its goals but I would say that the main conclusion, while plausible, are not entirely convincing.

Major

- The authors have, for a substantial part of their paper, a strong quantitative approach. This is less so for distinguishing between hypothesis a and b. Would it be possible to specify the outcomes of each of these hypothesis in a quantitative way (for instance via an RDM or other type of prediction). This would make it possible to see what is the better model and to what degree. With the current setup the main conclusions are somewhat narrative.

Thanks for this suggestion. We agree that the introduction did not lay out all of the predictions of the two accounts as clearly as it could. We have now revised the manuscript such that the predictions of each account are clearer and can be used to guide the reader throughout the manuscript. We now make concrete predictions in the main text with respect to each of the different theoretical accounts.

- o Currently the FC1 model is trained with data from all age ranges. What happens if you would train with age range 2-4 and age range 8-10? If I understand the reasoning this is not expected to be symmetrical? I think the authors would predict 8-10 would classify, on average 2-4 better, than vice versa. Is this correct? Hypothesis b would not predict this?

We too were curious about this idea, and indeed the reviewer's prediction does hold true: the feature representations in drawings from 8-10 year-olds can be used to classify the drawings made by 2-4 year olds, but the inverse is quite poor. However, we do not think that this analysis distinguishes between our hypotheses about the sources of developmental changes and so we have omitted it from the manuscript for brevity.

- o As an example of a narrative conclusion, on page 12 the authors state that the children's ability to control and plan their motor movements, but does not entirely account for. To what degree is does and does not?

Thanks, this suggestion is helpful in leading us to think about how to compare the strength of different predictors. In the revised manuscript, we have now standardized (z-scored) our predictors so that the coefficient estimates are on the same scale and can be compared in their magnitudes; these revised statistics can now be seen in Table 1.

- Much depends on the assumptions that FC1 of VGG can be used to evaluate the amount of diagnostic features in an image. I think it is therefore paramount that the authors show that a similar pattern is not present for Conv1 and Conv2, as a control. It would also be interesting to see to what degree this develops over the layers, and how/if this relates to development.

We took several steps to address this concern. First, we ran the suggested controls with Conv1/Conv2 (as well as all of the additional layers in VGG-19); the steep age-related change were only present when using activations from higher-level layers, suggesting that these developmental changes reflect the inclusion of mid- to high-level visual features (see Appendix, Figure A5).

To validate our main results with a different model, we also re-ran these same analyses using a large neural network model (CLIP, Contrastive-Language-Image Pairing model, Radford et al., 2021) that is now state of the art for zero-shot recognition of novel visual concepts. We found

the same results using the embeddings from this additional model; these results are now detailed throughout the manuscript and graphically in the Appendix.

Minor

• p8, it seems like a weak control to ask parents how often children draw each category, although better than alternatives. Would it be also possible to relate this to a popular source of frequencies of words? This might tap into the exposure to a certain category.

Thanks for this suggestion, which we have added to the revised manuscript. We do not find that word frequencies are a strong predictor of drawing recognizability.

• A change that is not covered by the study, but is referenced to, is the context of an object in relation to other objects. This is not covered by the drawings of single objects. Any gains on this basis cannot be evaluated.

Thanks for this comment. We didn't mean to imply that we had any insight into the development of context-based visual recognition and have checked that we don't make any claims about this.

Disclaimer: I have no specific background in developmental studies, mainly vision and DNNs

Reviewers' Comments:

Reviewer #2:

Remarks to the Author:

This is a revised manuscript, which was previously declined for publication. To their credit, the authors have addressed some, but not all, of the methodological issues that were raised during the prior review of the manuscript, with additional experimental work and analyses with a new sample of children. The manuscript is also well-written. While this new follow-up submission does address some of the methodological limitations noted previously (i.e., replicating main findings under tighter experimental conditions without the possibility of parental intervention), my previous concerns about the paper's substantive contribution remain.

The virtues of the paper continue to be in the collection and use of a very large dataset of children's drawings and visual recognition performance in a museum setting, and the sophisticated machine learning and computational modeling analyses that have been applied to this dataset. But the hypotheses and conclusions largely come off as being obvious, and in many cases the alternative hypotheses themselves seem like "straw people." In related instances, the evidence reported as tests of some of the hypotheses is more circumstantial than conclusive.

In my view, the greatest strength of the paper is its value as a proof-of-concept report. It demonstrates the possibility of large-scale data collection from a sample of children across different ages in a widely visited museum setting, and the application of machine learning techniques afforded by such a large-scale dataset from children. What is shown, however, is not all that novel. Because the advances in the paper are more methodological than substantive, I think the paper is better suited for a more discipline-specific journal, rather than such a general and selective outlet like *Nature Communications*.

Some of the key hypotheses in the paper seem obvious. A general hypothesis raised in the Introduction (and Discussion) is that children's visual concepts, reflected through production and recognition tasks, develop gradually. A related "straw person" hypothesis that is put forth in the Introduction is that the development of children's visual production and recognition abilities is largely complete by the preschool years. Likewise, the hypothesis that visuomotor development (operationalized here by tracing) primarily accounts for advances in visual production skills is simplistic and comes off also as a "straw person" hypothesis. Most theories of drawing do not attribute advances in drawing to visuomotor development alone. Another obvious conclusion is that children's drawings of visual concepts become more recognizable across childhood. That said, the machine learning procedures that are used to test or demonstrate these mostly obvious hypotheses and conclusions are sophisticated, and have not been previously employed in the developmental literature on drawing.

The interesting issue that is raised in the paper concerns the relation between the development of visual production and visual recognition skills. But as the authors acknowledge (in a footnote), the findings are indirect, and in essence are correlational. There is little direct empirical consideration or direct testing of the mechanisms or different models that may lead production and recognition skills to develop roughly in parallel over developmental time.

The authors also do not consider the visual statistics of children's visual world in order to interpret their findings on the developmental relationship between visual production and recognition skills. For instance, the classification accuracy of children's drawings of an infrequently drawn category (mushrooms) was more accurate than a commonly drawn category (dog). But to go beneath the surface of this finding, information is also needed about the variability of exposure to exemplars of these different categories, not just relative frequency of exposure.

Overall, then, I think the paper demonstrates the potential of a machine learning approach to understand the development of children's drawings. However, on this front, the conclusions are not that surprising. Additionally, the paper does not clearly establish how visual production and visual recognition skills are developmentally related at a mechanistic level other than demonstrating an association. For these reasons, I think this work is better suited for a more discipline-specific journal.

Reviewer #3:

Remarks to the Author:

The authors addressed my point of critique adequately and I have no further comments - congratulations from my side!

Reviewer #4:

Remarks to the Author:

I want to thank to authors for the extensive responses to the reviewers. Apart from the responses to my own queries I found both the questions and responses to reviewer 1 informative and convincing.

The flow of the manuscript has much improved.

Table 1 helps.

Also, I am happy to see the results are not also present in the early convolutional layers. Also very timely to have included CLIP.

All in all I think this is a great example of how specific (medium large) datasets in combination with modern AI shed a novel light on, in this case, developmental psychology.

Reviewer #5:

Remarks to the Author:

This manuscript presents an impressive dataset of childrens' drawing of visual object categories across a wide age range of childhood and examines how those drawings change over time. The unique dataset is complemented by a series of sophisticated analyses using both deep neural networks and observer ratings to tease apart aspects of that development. Yet, despite the quality of the dataset and the analyses, it's hard to tease apart exactly what we learn. I did not see the prior version, but the authors have made significant attempts to address the concerns raised by the reviewers. However, on my reading of the revised manuscript, I still had many of the same concerns as Reviewers 1 and 2. I think part of those concerns reflects a lack of clarity on the part of the authors on what they are actually investigating. The term 'visual concepts' is used a lot, but the authors never clearly define what they mean by that. I assume that what they mean is 'how something looks'. Drawings are a clever way to try and access those visual concepts, but there are clearly limitations. Those drawings will also reflect drawing skill, motor skill, limitations of the drawing tool and an understanding of how things are typically represented. Some aspects of visual concepts will rarely be reflected in the types of drawing examined here e.g. color or visual texture. In the manuscript it feels as if the authors are often analysing and interpreting the drawings as if they *are* the visual concept itself and not just a reflection of that concept through one possible form of output. The challenge for the current study is trying to understand what factors affect the nature of the drawings – is it a change in the visual concept or is it a change in something else such as understanding how things are typically represented or motor skill. To their credit, the authors run analyses to address exactly these concerns using a control task to measure skill and a recognition task. But the basic framework and limitation of this type of study of visual concepts is not clearly and explicitly laid out – personally, I think the manuscript would be a lot stronger if these sorts of consideration were clearly laid out in the introduction. For example, what are the advantages and disadvantages of using drawings as a way to assess visual concepts as compared to say verbal report? What aspects of visual concepts are not likely to be reflected in the drawings collected?

Despite my concerns, I think this is a potentially important and influential manuscript and my concerns could be alleviated with some rewriting and reframing. My suggestions would be to 1) clearly define what is meant by a visual concept, 2) lay out why drawings are a good way to

assess those visual concepts, 3) lay out what are the limitations of using drawings, and 4) clearly distinguish between what the authors are trying to measure (i.e. visual concepts) and what output they are measuring (i.e. drawings) throughout the manuscript – the authors are investigating how drawings change and inferring how visual concepts are developing. That inference is challenging, but the data presented here are extremely valuable – even just the descriptive aspects of this work (how the drawings change) are important. The framing of the related manuscript included with this submission is a lot better and more informative in my opinion.

REVIEWER COMMENTS

Reviewer #2 (Remarks to the Author):

This is a revised manuscript, which was previously declined for publication. To their credit, the authors have addressed some, but not all, of the methodological issues that were raised during the prior review of the manuscript, with additional experimental work and analyses with a new sample of children. The manuscript is also well-written. While this new follow-up submission does address some of the methodological limitations noted previously (i.e, replicating main findings under tighter experimental conditions without the possibility of parental intervention), my previous concerns about the paper's substantive contribution remain.

Thank you for continuing to serve as a reviewer on this paper.

The virtues of the paper continue to be in the collection and use of a very large dataset of children's drawings and visual recognition performance in a museum setting, and the sophisticated machine learning and computational modeling analyses that have been applied to this dataset. But the hypotheses and conclusions largely come off as being obvious, and in many cases the alternative hypotheses themselves seem like "straw people." In related instances, the evidence reported as tests of some of the hypotheses is more circumstantial than conclusive.

In my view, the greatest strength of the paper is its value as a proof-of-concept report. It demonstrates the possibility of large-scale data collection from a sample of children across different ages in a widely visited museum setting, and the application of machine learning techniques afforded by such a large-scale dataset from children. What is shown, however, is not all that novel. Because the advances in the paper are more methodological than substantive, I think the paper is better suited for a more discipline-specific journal, rather than such a general and selective outlet like Nature Communications.

Some of the key hypotheses in the paper seem obvious. A general hypothesis raised in the Introduction (and Discussion) is that children's visual concepts, reflected through production and recognition tasks, develop gradually. A related "straw person" hypothesis that is put forth in the Introduction is that the development of children's visual production and recognition abilities is largely complete by the preschool years. Likewise, the hypothesis that visuomotor development (operationalized here by tracing) primarily accounts for advances in visual production skills is simplistic and comes off also as a "straw person" hypothesis. Most theories of drawing do not attribute advances in drawing to visuomotor development alone. Another obvious conclusion is that children's drawings of visual concepts become more recognizable across childhood. That said, the machine learning procedures that are used to test or demonstrate these mostly obvious hypotheses and conclusions are sophisticated, and have not been previously employed in the developmental literature on drawing.

We appreciate your engagement with our work and ideas and for suggesting a more appropriate framing for the paper. While both of these "straw person" hypotheses have indeed been raised in many theoretical and empirical papers (including a recent review), we agree that they are overly simplistic. We have now significantly revised the Introduction to make the main contributions of our paper clear and clarified our theoretical viewpoint. Our introduction now motivated the present study without this overly simplistic, hypothesis testing framework and we have added a section highlighting the main methodological contributions of the paper at the end of the Introduction.

We stand by the view that this paper (1) provides evidence that children's drawings reflect their internal visual concepts for common object categories and (2) gives a methodological framework for testing new hypotheses about how changes in internal visual concepts might manifest in visual production and recognition. We believe that the revised manuscript now more clearly highlights these contributions, stated explicitly at the end of the introduction.

The interesting issue that is raised in the paper concerns the relation between the development of visual production and visual recognition skills. But as the authors acknowledge (in a footnote), the findings are indirect, and in essence are correlational. There is little direct empirical consideration or direct testing of the mechanisms or different models that may lead production and recognition skills to develop roughly in parallel over developmental time.

We agree that these results are indeed correlational, and we are enthusiastic about future work that empirically relates visual production and recognition skills in children to directly test different mechanisms (several studies that relate these two skills are cited in the Introduction). However, our present data are indeed limited and cannot fully distinguish those finer-grained hypotheses. In our revision, we tried to make this clear in three ways, via (1) content to the Introduction highlighting this work as a first step in this domain and highlighting our contributions as above, (2) the footnote (relevant to the specific correlational analyses) and (3) our limitations section (see the last paragraph of the GD) where we call for future work and within-child studies to validate and confirm these findings.

The authors also do not consider the visual statistics of children's visual world in order to interpret their findings on the developmental relationship between visual production and recognition skills. For instance, the classification accuracy of children's drawings of an infrequently drawn category (mushrooms) was more accurate than a commonly drawn category (dog). But to go beneath the surface of this finding, information is also needed about the variability of exposure to exemplars of these different categories, not just relative frequency of exposure.

The statistics of the visual world almost certainly impact these item effects in a variety of ways. Perhaps as you are intuiting, it seems plausible that *mushrooms* have less shape variability (particularly in depictions) than do *dogs*, and that this would make them simply easier to draw recognizable versions of. We note that our current measures do not take into account frequency of exposure (rather only measuring word frequency) and we suspect that a combination of the frequency of visual experience, the diversity of this experience across visual formats (e.g., drawings, illustrations, realistic photographs) impacts children's internal visual concepts. As even this short discussion makes clear, measuring all of these factors is a massive undertaking given that standardized metrics and instruments for each simply don't exist. In the revised General Discussion, we now highlight this interesting idea as an avenue for future work.

Overall, then, I think the paper demonstrates the potential of a machine learning approach to understand the development of children's drawings. However, on this front, the conclusions are not that surprising. Additionally, the paper does not clearly establish how visual production and visual recognition skills are developmentally related at a mechanistic level other than demonstrating an association. For these reasons, I think this work is better suited for a more discipline-specific journal.

We respectfully disagree. We believe that this paper would be beneficial to a broader scientific audience. In our experience, this project has engaged scholars across a much wider breadth of disciplines than most other ongoing projects in our labs, and led to many new ideas for future work based on these findings and novel data collection and analysis techniques. The paper highlights a novel approach to understanding changes in children's visual concepts, which has mostly only been examined via standard recognition tasks with realistic photographic exemplars. We hope that the revised manuscript now clarifies these contributions.

Reviewer #3 (Remarks to the Author):

The authors addressed my point of critique adequately and I have no further comments - congratulations from my side!

Thank you!

Reviewer #4 (Remarks to the Author):

I want to thank to authors for the extensive responses to the reviewers. Apart from the responses to my own queries I found both the questions and responses to reviewer 1 informative and convincing.

The flow of the manuscript has much improved.
Table 1 helps.

Also, I am happy to see the results are not also present in the early convolutional layers. Also very timely to have included CLIP.

All in all I think this is a great example of how specific (medium large) datasets in combination with modern AI shed a novel light on, in this case, developmental psychology.

Thank you!

Reviewer #5 (Remarks to the Author):

This manuscript presents an impressive dataset of childrens' drawing of visual object categories across a wide age range of childhood and examines how those drawings change over time. The unique dataset is complemented by a series of sophisticated analyses using both deep neural networks and observer ratings to tease apart aspects of that development. Yet, despite the quality of the dataset and the analyses, it's hard to tease apart exactly what we learn. I did not see the prior version, but the authors have made significant attempts to address the concerns raised by the reviewers.

Thank you for engaging with our work and for serving as a reviewer.

However, on my reading of the revised manuscript, I still had many of the same concerns as Reviewers 1 and 2. I think part of those concerns reflects a lack of clarity on the part of the authors on what they are actually investigating. The term 'visual concepts' is used a lot, but the authors never clearly define what they mean by that. I assume that what they mean is 'how something looks'.

Thank you for your feedback. In the revised manuscript, we have clarified what we mean by visual concept”—our representations of “what things look like.”—by revising the Introduction to the paper.

Drawings are a clever way to try and access those visual concepts, but there are clearly limitations. Those drawings will also reflect drawing skill, motor skill, limitations of the drawing tool and an understanding of how things are typically represented. Some aspects of visual concepts will rarely be reflected in the types of drawing examined here e.g. color or visual texture. In the manuscript it feels as if the authors are often analysing and interpreting the drawings as if they *are* the visual concept itself and not just a reflection of that concept through one possible form of output. The challenge for the current study is trying to understand what factors affect the nature of the drawings – is it a change in the visual concept or is it a change in something else such as understanding how things are typically represented or motor skill.

Thank you for your engagement with the paper. We have restructured the paper to highlight this central challenge, in particular, whether changes in drawings reflect changes in children’s internal visual concepts. Our prior version of the paper was grounded in hypotheses from literature (e.g., visual concepts mature early vs. change continually), but we agree with your comments and the previous reviewer that these in fact feel like “straw person” hypotheses that are ultimately disconnected from the data at hand. This has resulted in major changes to the structure of the Introduction and General Discussion.

To their credit, the authors run analyses to address exactly these concerns using a control task to measure skill and a recognition task. But the basic framework and limitation of this type of study of visual concepts is not clearly and explicitly laid out – personally, I think the manuscript would be a lot stronger if these sorts of consideration were clearly laid out in the introduction. For example, what are the advantages and disadvantages of using drawings as a way to assess visual concepts as compared to say verbal report? What aspects of visual concepts are not likely to be reflected in the drawings collected?

We agree that drawings are only one of many tools that can be used to assess changes in visual concepts, and your review highlights that our manuscript did not do an adequate job of conveying the pros and cons of using drawings to study these internal representations. In the revised manuscript, we have added several paragraphs to address this issue. We now motivate the use of drawings as a method to investigate visual concepts in the Introduction (see pgs. 4-5), and raise directions for future work (specifically with regards to item variation) in the General Discussion (see pg. 26).

Despite my concerns, I think this is a potentially important and influential manuscript and my concerns could be alleviated with some rewriting and reframing. My suggestions would be to 1) clearly define what is meant by a visual concept, 2) lay out why drawings are a good way to assess those visual concepts, 3) lay out what are the limitations of using drawings, and 4) clearly distinguish between what the authors are trying to measure (i.e. visual concepts) and what output they are measuring (i.e. drawings) throughout the manuscript – the authors are investigating how drawings change and inferring how visual concepts are developing. That inference is challenging, but the data presented here are extremely valuable – even just the descriptive aspects of this work (how the drawings change) are important. The framing of the related manuscript included with this submission is a lot better and more informative in my opinion.

We appreciate your concrete and helpful suggestions for the manuscript. We have significantly revised the manuscript in accordance with your suggestions, and we believe that the manuscript is now much improved.

Reviewers' Comments:

Reviewer #2:

Remarks to the Author:

Overall, I believe the authors have made a conscientious attempt to respond to the comments in the prior set of reviews. The authors make a convincing case for their innovative approach for investigating the development of children's drawings, including the use of data collection procedures in a museum setting affording an extremely large number of participants (over 37k) and the use of machine learning analytic procedures. My view is that this paper substantially advances the literature on children's drawings, and links up meaningfully with the literatures on category learning and visual categorization. My remaining comments have to do with clarifications and future directions for research based on the authors' report.

General comments

Some of the analyses span the entire age range (2-10 years), and other analyses appear to omit 2-year-olds, and span the 3- to 10-year-old period. The authors may want to comment on why this was done, and whether this affects the ability to make comparisons across the different sets of findings that are reported.

In several cases throughout the results, especially with respect to the younger children in the sample, the findings exceed chance levels, but children are nevertheless quite inaccurate. In these instances, some qualifications should be made either when the findings are presented and/or in the General Discussion.

Other

p. 6, line, 110: "To quantify...

Figure 2B: Should a legend be provided to indicate what color corresponds to which age?

Figure 5c: Does the assumption always hold that the size of a drawn feature corresponds to the importance of the feature? While I can see how this often would be the case, I can imagine other instances where a small feature is nevertheless important for diagnostic or identification purposes. There also seems to be an assumption here that children will adjust the scale of the feature relative to the drawing to emphasize the importance of a diagnostic feature.

p. 19: The authors raise the question of why do children include more diagnostic information in their drawings with increasing age. Another possibility that might be considered (albeit not mutually exclusive with those already mentioned in the manuscript) is that children are becoming less egocentric and more aware that their drawings potentially have a communicative function. In other words, we might also be seeing some type of audience effect with increasing age. I don't think this possibility undercuts the authors' arguments, but perhaps it should be mentioned.

p. 20: As a follow-up validation study, but not for this paper, the authors may want to consider whether drawings where a diagnostic feature is allotted more space (information contained in Figure 5c) is more easily recognized in the Guessing Task. This type of analysis may also be relevant for interpreting the unexpected interaction report on p. 22, where drawings with more diagnostic features were less well recognized than drawings with an intermediate number of diagnostic features at older age levels.

p. 26: The authors may want to refer to the work of Lisa Oakes, who has found differences in infants' visual categorization of household animals, based on whether their households actually contain an animal pet.

p. 32: Method: Out of 11797 subsequent sessions at the station, 3094 filled out the survey, and 719 reported interference, 6.09% of participants I still think this statement is misleading, since the percentage should be based on the number who filled out the survey. If based on those who filled out the survey, the proportion of sessions where interference was observed would be closer to

23%, which is not insignificant. It would be helpful to report the age breakdown for the 719 cases of observed interference and let readers draw their own conclusions.

p. 33: I may have missed it, but in the Method section, report the age breakdown of the children in the supplemental experiment.

Reviewer #5:

Remarks to the Author:

The authors have thoughtfully and carefully revised the manuscript in response to the concerns raised and I think it is now in excellent shape. This is a very nice study and I have no further comments.

REVIEWERS' COMMENTS

Reviewer #2 (Remarks to the Author):

Overall, I believe the authors have made a conscientious attempt to respond to the comments in the prior set of reviews. The authors make a convincing case for their innovative approach for investigating the development of children's drawings, including the use of data collection procedures in a museum setting affording an extremely large number of participants (over 37k) and the use of machine learning analytic procedures. My view is that this paper substantially advances the literature on children's drawings, and links up meaningfully with the literatures on category learning and visual categorization. My remaining comments have to do with clarifications and future directions for research based on the authors' report.

Thank you for serving as a reviewer and for your comments on this work!

General comments

Some of the analyses span the entire age range (2-10 years), and other analyses appear to omit 2-year-olds, and span the 3- to 10-year-old period. The authors may want to comment on why this was done, and whether this affects the ability to make comparisons across the different sets of findings that are reported.

Thanks for your attention to detail. We mentioned this in the Methods section, but we were concerned that the guessing games were eliciting a high amount of interference for the 2-year-olds (for whom a 4AFC task is admittedly quite challenging even under scaffolded conditions), and decided to omit them from these analyses (this choice was pre-registered after examining initial data). We have now added this reasoning to the main text.

In several cases throughout the results, especially with respect to the younger children in the sample, the findings exceed chance levels, but children are nevertheless quite inaccurate. In these instances, some qualifications should be made either when the findings are presented and/or in the General Discussion.

We have added additional qualifications in several places, acknowledging the item variation we see in the main study (Figures 2A,B), and in animacy/real-world size classifications, and in the drawing recognition data (where 3-year-olds are near chance).

Other

p. 6, line, 110: "To quantify..."

Thanks for catching this typo.

Figure 2B: Should a legend be provided to indicate what color corresponds to which age?

Thanks for catching this. We have amended the figure caption such that the reader can refer to Figure 2A, as these are the same colors; we have also modified the caption for Figure 6.

Figure 5c: Does the assumption always hold that the size of a drawn feature corresponds to the importance of the feature? While I can see how this often would be the case, I can imagine other instances where a small feature is nevertheless important for diagnostic or identification purposes. There also seems to be an

assumption here that children will adjust the scale of the feature relative to the drawing to emphasize the importance of a diagnostic feature.

Thanks for this question, as it reveals that we were not sufficiently clear about this analysis. The data in Figure 5c show that children’s drawings of inanimate objects might not be recognizable as a “piano” or a “bench” but still contain visual information that allows something to be classified as “a something that is big in the real-world” (and vice versa for small, inanimate objects) because big objects tend to have boxier shape structures. We have clarified this entire section in the paper.

p. 19: The authors raise the question of why do children include more diagnostic information in their drawings with increasing age. Another possibility that might be considered (albeit not mutually exclusive with those already mentioned in the manuscript) is that children are becoming less egocentric and more aware that their drawings potentially have a communicative function. In other words, we might also be seeing some type of audience effect with increasing age. I don’t think this possibility undercuts the authors’ arguments, but perhaps it should be mentioned.

We agree that this is a real possibility, and now highlight it in the Discussion in the section under “possible mechanisms”.

p. 20: As a follow-up validation study, but not for this paper, the authors may want to consider whether drawings where a diagnostic feature is allotted more space (information contained in Figure 5c) is more easily recognized in the Guessing Task. This type of analysis may also be relevant for interpreting the unexpected interaction report on p. 22, where drawings with more diagnostic features were less well recognized than drawings with an intermediate number of diagnostic features at older age levels.

Thanks for this suggestion! This is an interesting idea for future studies.

p. 26: The authors may want to refer to the work of Lisa Oakes, who has found differences in infants’ visual categorization of household animals, based on whether their households actually contain an animal pet.

We agree this is a relevant reference and have added it to the Introduction introducing these topics.

p. 32: Method: Out of 11797 subsequent sessions at the station, 3094 filled out the survey, and 719 reported interference, 6.09% of participants I still think this statement is misleading, since the percentage should be based on the number who filled out the survey. If based on those who filled out the survey, the proportion of sessions where interference was observed would be closer to 23%, which is not insignificant. It would be helpful to report the age breakdown for the 719 cases of observed interference and let readers draw their own conclusions.

We understand the concern. We have omitted the “6.09%” statistic from this section and added the age breakdown by interference and interference type (parent vs. other child) as a table in the SI.

p. 33: I may have missed it, but in the Method section, report the age breakdown of the children in the supplemental experiment.

We have added this information; approximately 20 participants were recruited in each age

group.

Reviewer #5 (Remarks to the Author):

The authors have thoughtfully and carefully revised the manuscript in response to the concerns raised and I think it is now in excellent shape. This is a very nice study and I have no further comments.

Thank you for serving as a reviewer!